# VALUE MEMORY GRAPH: A GRAPH-STRUCTURED WORLD MODEL FOR OFFLINE REINFORCEMENT LEARNING

**Deyao Zhu**[1] **Li Erran Li**[2]* **Mohamed Elhoseiny**[1]

[1] King Abdullah University of Science and Technology

[2] AWS AI, Amazon and Columbia University

`{deyao.zhu, mohamed.elhoseiny}@kaust.edu.sa, erranlli@gmail.com`

## ABSTRACT

Reinforcement Learning (RL) methods are typically applied directly in environments to learn policies. In some complex environments with continuous state-action spaces, sparse rewards, and/or long temporal horizons, learning a good policy in the original environments can be difficult. Focusing on the offline RL setting, we aim to build a simple and discrete world model that abstracts the original environment. RL methods are applied to our world model instead of the environment data for simplified policy learning. Our world model, dubbed **V**alue **M**emory **G**raph (VMG), is designed as a directed-graph-based Markov decision process (MDP) of which vertices and directed edges represent graph states and graph actions, separately. As state-action spaces of VMG are finite and relatively small compared to the original environment, we can directly apply the value iteration algorithm on VMG to estimate graph state values and figure out the best graph actions. VMG is trained from and built on the offline RL dataset. Together with an action translator that converts the abstract graph actions in VMG to real actions in the original environment, VMG controls agents to maximize episode returns. Our experiments on the D4RL benchmark show that VMG can outperform state-of-the-art offline RL methods in several goal-oriented tasks, especially when environments have sparse rewards and long temporal horizons. Code is available at `https://github.com/TsuTikgiau/ValueMemoryGraph`

## 1 INTRODUCTION

Humans are usually good at simplifying difficult problems into easier ones by ignoring trivial details and focusing on important information for decision making. Typically, reinforcement learning (RL) methods are directly applied in the original environment to learn a policy. When we have a difficult environment like robotics or video games with long temporal horizons, sparse reward signals, or large and continuous state-action space, it becomes more challenging for RL methods to reason the value of states or actions in the original environment to get a well-performing policy. Learning a world model that simplifies the original complex environment into an easy version might lower the difficulty to learn a policy and lead to better performance.

In offline reinforcement learning, algorithms can access a dataset consisting of pre-collected episodes to learn a policy without interacting with the environment. Usually, the offline dataset is used as a replay buffer to train a policy in an off-policy way with additional constraints to avoid distribution shift problems (Wu et al., 2019; Fujimoto et al., 2019; Kumar et al., 2019; Nair et al., 2020; Wang et al., 2020; Peng et al., 2019). As the episodes also contain the dynamics information of the original environment, it is possible to utilize such a dataset to directly learn an abstraction of the environment in the offline RL setting. To this end, we introduce **V**alue **M**emory **G**raph (**VMG**), a graph-structured world model for offline reinforcement learning tasks. VMG is a Markov decision process (MDP) defined on a graph as an abstract of the original environment. Instead of directly applying RL methods to the offline dataset collected in the original environment, we learn and build VMG first and use

---

*Work done outside of Amazon

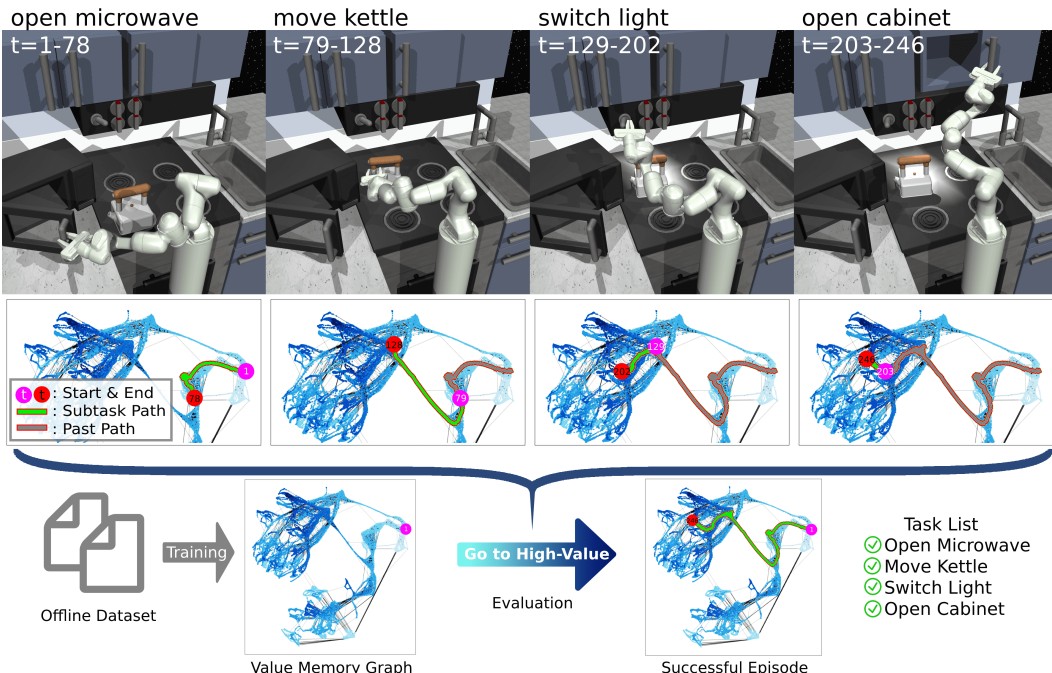

Figure 1: Demonstration of a successful episode where a robot trained in the dataset "kitchen-partial" accomplishes 4 subtasks in sequence guided by VMG. Vertex values are shown via color shade. By searching graph actions that lead to the high-value future region (darker blue) calculated by value iteration on the graph, VMG controls the robot arm to maximize episode rewards and finish the task.

it as a simplified substitute of the environment to apply RL methods. VMG is built by mapping offline episodes to directed chains in a metric space trained via contrastive learning. Then, these chains are connected to a graph via state merging. Vertices and directed edges of VMG are viewed as graph states and graph actions. Each vertex transition on VMG has rewards defined from the original rewards in the environment.

To control agents in environments, we first run the classical value iteration algorithm(Puterman, 2014) once on VMG to calculate graph state values. This can be done in less than one second without training a value neural network thanks to the discrete and relatively smaller state and action spaces in VMG. At each timestep, VMG is used to search for graph actions that can lead to high-value future states. Graph actions are directed edges and cannot be directly executed in the original environment. With the help of an action translator trained in supervised learning (e.g., Emmons et al. (2021)) using the same offline dataset, the searched graph actions are converted to environment actions to control the agent. An overview of our method is shown in Fig.1.

Our contribution can be summarized as follows:

- We present **V**alue **M**emory **G**raph (VMG), a graph-structured world model in offline reinforcement learning setting. VMG represents the original environments as a graph-based MDP with relatively small and discrete action and state spaces.

- We design a method to learn and build VMG on an offline dataset via contrastive learning and state merging.

- We introduce a VMG-based method to control agents by reasoning graph actions that lead to high-value future states via value iteration and convert them to environment actions via an action translator.

- Experiments on the D4RL benchmark show that VMG can outperform several state-of-the-art offline RL methods on several goal-oriented tasks with sparse rewards and long temporal horizons.

## 2 RELATED WORK

**Offline Reinforcement Learning**  One crucial problem in offline RL is how to avoid out-of-the-training-distribution (OOD) actions and states that decrease the performance in test environments (Fujimoto et al., 2019; Kumar et al., 2019; Levine et al., 2020). Recent works like Wu et al. (2019); Fujimoto et al. (2019); Kumar et al. (2019); Nair et al. (2020); Wang et al. (2020); Peng et al. (2019) directly penalize the mismatch between the trained policy and the behavior policy via an explicit density model or via divergence. Another methods like Kumar et al. (2020); Kostrikov et al. (2021a;b) constrains the training via penalizing the Q function. Model-based reinforcement learning methods like Yu et al. (2020; 2021); Kidambi et al. (2020) constrains the policy to the region of the world model that is close to the training data. Compared to previous methods, graph actions in VMG always control agents to move to graph states, and all the graph states come from the training dataset. Therefore, agents stay close to states from the training distribution naturally.

**Hierarchical Reinforcement Learning**  Hierarchical RL methods (e.g., Savinov et al. (2018); Nachum et al. (2018); Eysenbach et al. (2019); Huang et al. (2019); Liu et al. (2020); Mandlekar et al. (2020); Yang et al. (2020); Emmons et al. (2020); Zhang et al. (2021)) use hierarchical policies to control agents with a high-level policy that generate commands like abstract actions or skills, and a low-level policy that converts them to concrete environment actions. Our method can be viewed as a hierarchical RL approach with VMG-based high-level policy and a low-level action translator. Compared to previous methods which learn high-level policies in environments, our high-level policy is instead trained in VMG via value iteration without additional neural network learning.

**Model-based Reinforcement Learning**  Recent research in model-based reinforcement learning (MBRL) has shown a significant advantage (Ha & Schmidhuber, 2018; Janner et al., 2019; Hafner et al., 2019; Schrittwieser et al., 2020; Ye et al., 2021) in sample efficiency over model-free reinforcement learning. In most of the previous methods, world models are designed to approximate the original environment transition. In contrast, VMG abstracts the environment as a simple graph-based MDP. Therefore, we can apply RL methods directly to VMG for simple and fast policy learning. As we demonstrate later in our experiments, this facilitates reasoning and leads to good performance in tasks with long temporal horizons and sparse rewards.

**Graph from Experience**  Similar to VMG, methods like Hong et al. (2022); Jiang et al. (2022); Shrestha et al. (2020); Marklund et al. (2020); Char et al. (2022) study the credit assignment problem on a graph created from the experience. Hong et al. (2022); Jiang et al. (2022) are designed for discrete environments. Shrestha et al. (2020); Marklund et al. (2020) considers the environments with finite actions and continuous state space by discretizing states or state features via kNN. Char et al. (2022) introduces a stitch operator to create a graph directly by adding new transitions. It can work with environments with low-dimensional continuous action spaces like Mountain Car Continuous (1 dimension) and Maze2D (2 dimensions). However, the stitch operator is hard to scale to high-dimensional action spaces. In contrast, VMG discretizes both state and action spaces and thus can work with continuous high-dimensional action spaces.

**Representation Learning**  Contrastive learning methods learns a good representation by maximizing the similarity between related data and minimizing the similarity of unrelated data (Oord et al., 2018; Chen et al., 2020; Radford et al., 2021) in the learned representation space. Bisimulation-based methods like Zhang et al. (2020) learn a representation with the help of bisimulation metrics (Ferns & Precup, 2014; Ferns et al., 2011; Bertsekas & Tsitsiklis, 1995) measuring the 'behavior similarity' of states w.r.t. future reward sequences given any input action sequences. In VMG, we use a contrastive learning loss to learn a metric space encoding the similarity between states as L2 distance.

## 3 VALUE MEMORY GRAPH (VMG)

Our world model, Value Memory Graph (VMG), is a graph-structured Markov decision process constructed as a simplified version of the original environment with discrete and relatively smaller state-action spaces. RL methods can be applied on the VMG instead of the original environment to lower the difficulty of policy learning. To build VMG, we first learn a metric space that measures the

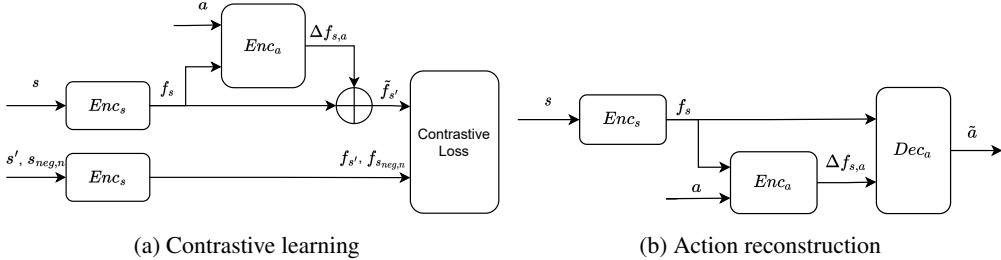

(a) Contrastive learning         (b) Action reconstruction

Figure 2: The training pipeline of the state encoder $Enc_s$ and the action encoder $Enc_a$ to build the memory map. $Enc_s$ converts original states $s$ into points in the memory map. $Enc_a$ maps actions $a$ as transitions in the memory map.

reachability among the environment states. Then, a graph is built in the metric space from the dataset as the backbone of our VMG. In the end, a Markov decision process is defined on the graph as an abstract representation of the environment.

## 3.1 VMG METRIC SPACE LEARNING

VMG is built in a metric space where the L2 distance represents whether one state can be reached from another state in a few timesteps. The embedding in the metric space is based on a contrastive-learning mechanism demonstrated in Fig.2a. We have two neural networks: a state encoder $Enc_s : s \rightarrow f_s$ that maps the original state $s$ to a state feature $f_s$ in the metric space, and an action encoder $Enc_a : f_s, a \rightarrow \Delta f_{s,a}$ that maps the original action $a$ to a transition $\Delta f_{s,a}$ in the metric space conditioned on the current state feature $f_s$. Given a transition triple $(s, a, s')$, we add the transition $\Delta f_{s,a}$ to the state feature $f_s$ as the prediction of the next state feature $\tilde{f}_{s'} = f_s + \Delta f_{s,a}$. The prediction is encouraged to be close to the ground truth $f_{s'}$ and away from other unrelated state features. Therefore, we use the following learning objective to train $Enc_s$ and $Enc_a$:

$$L_c = D^2(\tilde{f}_{s'}, f_{s'}) + \frac{1}{N} \sum \max(m - D^2(\tilde{f}_{s'}, f_{s_{neg,n}}), 0) \tag{1}$$

Here, $D(\cdot, \cdot)$ denotes the L2 distance. $s_{neg,n}$ denotes the $n$-th negative state. Given a batch of transition triples $(s_i, a_i, s'_i)$ randomly sampled from the training set and a fixed margin distance $m$, we use all the other next states $s'_{j|j \neq i}$ as the negative states for $s_i$ and encourage $\tilde{f}_{s'_i}$ to be at least $m$ away from negative states in the metric space. In addition, we use an action decoder $Dec_a : f_s, \Delta f_{s,a} \rightarrow \tilde{a}$ to reconstruct the action from the transition $\Delta f_{s,a}$ conditioned on the state feature $f_s$ as shown in Fig.2b. This conditioned auto-encoder structure encourages the transition $\Delta f_{s,a}$ to be a meaningful representation of the action. Besides, we penalize the length of the transition when it is larger than the margin $m$ to encourage adjacent states to be close in the metric space. Therefore, we have the additional action loss $L_a$ shown below.

$$L_a = D^2(\tilde{a}, a) + \max(\|\Delta f_{s,a}\|_2 - m, 0) \tag{2}$$

$L_{\text{metric}}$, the total training loss for metric learning, is the sum of the contrastive and action losses.

$$L_{\text{metric}} = L_c + L_a \tag{3}$$

## 3.2 CONSTRUCT THE GRAPH IN VMG

To construct the graph in VMG, we first map all the episodes in the training data to the metric space as directed chains. Then, these episode chains are combined into a graph with a reduced number of state features. This is done by merging similar state features into one vertex based on the distance in the metric space. The overall algorithm are visualized in Fig.3a and can be found in Appx.B.1. Given a distance threshold $\gamma_m$, a vertex set $\mathcal{V}$, and a checking state $s_i$, we check whether the minimal distance in the metric space from the existing vertices to the checking state $s_i$ is smaller than $\gamma_m$. If not or if the vertex set is empty, we set the checking state $s_i$ as a new vertex $v_J$ and add it to $\mathcal{V}$. This process is repeated over the whole dataset. After the vertex set $\mathcal{V}$ is constructed, each state $s_i$ can be classified into a vertex $v_j$ of which the distance in the metric space is smaller than $\gamma_m$. In the training

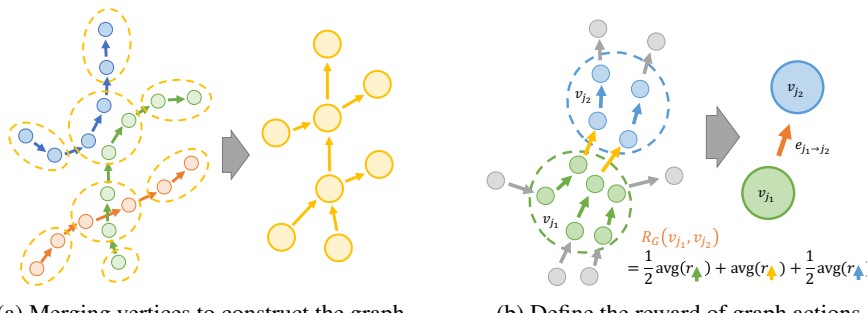

(a) Merging vertices to construct the graph  (b) Define the reward of graph actions

Figure 3: Create a graph and define rewards in VMG. In Fig.3a, three episodes are mapped as three chains in the metric space colored differently. We merge nodes that are close to each other together and combine these chains into a directed graph. In Fig.3b, the graph reward $R_G(v_{j_1}, v_{j_2})$ of the action from the green vertex $v_{j_1}$ to the blue vertex $v_{j_2}$ is defined as the average over rewards in the original episodes.

set, each state transition $(s_i, a_i, s_i')$ represents a directed connection from $s_i$ to $s_i'$. Therefore, we create the graph directed edges from the original transitions. For any two different vertices $v_{j_1}, v_{j_2}$ in $\mathcal{V}$, if there exist a transition $(s_i, a_i, s_i')$ where $s_i$ and $s_i'$ can be classified into $v_{j_1}$ and $v_{j_2}$, respectively, we add a directed edge $e_{j_1 \to j_2}$ from $v_{j_1}$ to $v_{j_2}$.

### 3.3  DEFINE A GRAPH-BASED MDP

VMG is a Markov decision process (MDP) $(\mathcal{S}_G, \mathcal{A}_G, \mathrm{P}_G, \mathrm{R}_G)$ defined on the graph. $\mathcal{S}_G, \mathcal{A}_G, \mathrm{P}_G, \mathrm{R}_G$ denotes the state set, the action set, the state transition probability, and the reward of this new graph MDP, respectively. Based on the graph, each vertex is viewed as a graph state. Besides, we view each directed connection $e_{j_1 \to j_2}$ starting from a vertex $v_{j_1}$ as an available graph action in $v_{j_1}$. Therefore, the graph state set $\mathcal{S}_G$ equals the graph vertex set $\mathcal{V}$ and the graph action set is the graph edge set $\mathcal{E}$. For the graph state transition probability $\mathrm{P}_G$ from $v_{j_1}$ to $v_{j_2}$, we define it as 1 if the corresponding edge exists in $\mathcal{E}$ otherwise 0. Therefore,

$$\mathrm{P}_G(v_{j_2}|v_{j_1}, e_{j_1 \to j_2}) = \begin{cases} 1 & \text{if } e_{j_1 \to j_2} \in \mathcal{E} \\ 0 & \text{otherwise} \end{cases} \tag{4}$$

We define the graph reward of each possible state transition $e_{j_1 \to j_2}$ as the average over the original rewards from $v_{j_1}$ to $v_{j_2}$ in the training set $\mathcal{D}$, plus "internal rewards". The internal reward comes from the original transitions that are inside $v_{j_1}$ or $v_{j_2}$ after state merging. An example of graph reward definition is visualized in Fig.3b. Concretely,

$$R_{j_1 \to j_2} = \text{avg}\{r_i | \forall s_i \text{ classified to } v_{j_1}, s_i' \text{ classified to } v_{j_2}, (s_i, a_i, r_i, s_i') \in \mathcal{D}\} \tag{5}$$

$$\mathrm{R}_G(v_{j_1}, v_{j_2}) = \begin{cases} \frac{1}{2}R_{j_1 \to j_1} + R_{j_1 \to j_2} + \frac{1}{2}R_{j_2 \to j_2} & \text{if } e_{j_1 \to j_2} \in \mathcal{E} \\ \text{Not defined} & \text{otherwise} \end{cases} \tag{6}$$

Note that the rewards of graph transitions outside of $\mathcal{E}$ are not defined, as these transitions will not happen according to Eq.4. For internal rewards where both the source $s_i$ and the target $s_i'$ of the original transition $(s_i, s_i')$ are classified to the same vertex, we split the reward into two and allocate them to both incoming and outgoing edges, respectively. This is shown as $\frac{1}{2}R_{j_1 \to j_1}$ and $\frac{1}{2}R_{j_2 \to j_2}$ in Eq.6. Now we have a well-defined MDP on the graph. This MDP serves as our world model VMG.

### 3.4  HOW TO USE VMG

VMG, together with an action translator, can generate environment actions that control agents to maximize episode returns. We first run the classical RL method value iteration (Puterman, 2014) on VMG to compute the value $V(v_j)$ of each graph state $v_j$. This can be done in one second without learning an additional neural-network-based value function due to VMG's finite and discrete state-action spaces.

To guide the agent, VMG provides a graph action that leads to high-value graph states in the future at each time step. Due to the distribution shift between the offline dataset and the environment, there

can be gaps between VMG and the environment. Therefore, the optimal graph action calculated directly by value iteration on VMG might not be optimal in the environment. We notice that instead of greedily selecting the graph actions with the highest next state values, searching for a good future state after multiple steps first and planning a path to it can give us a more reliable performance. Given the current environment state $s_c$, we first find the closest graph state $v_c$ on VMG. Starting from $v_c$, we search for $N_s$ future steps to find the future graph state $v^*$ with the best value. Then, we plan a shortest path $\mathcal{P} = [v_c, v_{c+1}, ..., v^*]$ from $v_c$ to $v^*$ via Dijkstra (Dijkstra et al., 1959) on the graph. We select the $N_{sg}$-th graph state $v_{c+N_{sg}}$ and make an edge $e_{c \rightarrow c+N_{sg}}$ as the searched graph action. The graph action $e_{c \rightarrow c+N_{sg}}$ is converted to the environment action $a_c$ via an action translator: $a_c = Tran(s_c, v_{c+N_{sg}})$. The pseudo algorithm can be found in Appx.B.2.

The action translator $Tran(s, s')$ reasons the executed environment action given the current state $s$ and a state $s'$ in the near future. $Tran(s, s')$ is trained purely in the offline dataset via supervised learning and separately from the training of VMG. In detail, given an episode from the training set and a time step $t$, we first randomly sample a step $t + k$ from the future $K$ steps. $k \sim Uniform(1, K)$. Then, $Tran(s, s')$ is trained to regress the action $a_t$ at step $t$ given the state $s_t$ and the future state $s_{t+k}$ using a L2 regression loss $L_{Tran} = D^2(Tran(s_t, s_{t+k}), a_t)$. Note that when $k = 1$, $p(a_t|s_t, s_{t+k})$ is determined purely by the environment dynamics and $Tran(s, s')$ becomes an inverse dynamics model. As $k$ increase, the influence of the behavior policy that collects the offline dataset on $p(a_t|s_t, s_{t+k})$ will increase. Therefore, the sample range $K$ should be small to reflect the environment dynamics and reduce the influence of the behavior policy. In all of our experiments, $K$ is set to 10.

# 4 EXPERIMENTS

## 4.1 PERFORMANCE ON OFFLINE RL BENCHMARKS

**Test Benchmark** We evaluate VMG on the widely used offline reinforcement learning benchmark D4RL (Fu et al., 2020). In detail, we test VMG on three domains: Kitchen, AntMaze, and Adorit. In Kitchen, a robot arm in a virtual kitchen needs to finish four subtasks in an episode. The robot receives a sparse reward after finishing each subtask. D4RL provides three different datasets in Kitchen: kitchen-complete, kitchen-partial, and kitchen-mixed. In AntMaze, a robot ant needs to go through a maze and reaches a target location. The robot only receives a sparse reward when it reaches the target. D4RL provides three mazes of different sizes. Each of them contains two datasets. In Adroit, policies control a robot hand to finish tasks like rotating a pen or opening a door with dense rewards. For evaluation, D4RL normalizes all the performance of different tasks to a range of 0-100, where 100 represents the performance of an "expert" policy. More benchmark details can be found in D4RL (Fu et al., 2020) and Appx.A.

Table 1: Experimental results on domains Kitchen, AntMaze, and Adroit from D4RL benchmark. VMG outperforms baselines in Kitchen and AntMaze where only sparse rewards are provided and achieves comparable performance in Adroit. Results and the standard deviation are calculated over three trained models.

| Dataset | BC | BRAC-p | BEAR | DT | AWAC | CQL | IQL | VMG |
|---|---|---|---|---|---|---|---|---|
| kitchen-complete | 65.0 | 0.0 | 0.0 | - | - | 43.8 | 62.5 | **73.0** $\pm$ 6.7 |
| kitchen-partial | 38.0 | 0.0 | 0.0 | - | - | 49.8 | 46.3 | **68.8** $\pm$ 11.9 |
| kitchen-mixed | **51.5** | 0.0 | 0.0 | - | - | 51.0 | 51.0 | 50.6 $\pm$ 4.1 |
| **kitchen-total** | 154.5 | 0.0 | 0.0 | - | - | 144.6 | 159.8 | **192.4** |
| antmaze-umaze | 54.6 | - | - | 59.2 | 56.7 | 74.0 | 87.5 | **93.7** $\pm$ 2.3 |
| antmaze-umaze-diverse | 45.6 | - | - | 53.0 | 49.3 | 84.0 | 62.2 | **94.0** $\pm$ 2.0 |
| antmaze-medium-play | 0.0 | - | - | 0.0 | 0.0 | 61.2 | 71.2 | **82.7** $\pm$ 3.1 |
| antmaze-medium-diverse | 0.0 | - | - | 0.0 | 0.7 | 53.7 | 70.0 | **84.3** $\pm$ 2.1 |
| antmaze-large-play | 0.0 | - | - | 0.0 | 0.0 | 15.8 | 39.6 | **67.3** $\pm$ 3.2 |
| antmaze-large-diverse | 0.0 | - | - | 0.0 | 1.0 | 14.9 | 47.5 | **74.3** $\pm$ 3.1 |
| **antmaze-total** | 100.2 | - | - | 112.2 | 107.7 | 303.6 | 378.0 | **496.3** |
| pen-human | 63.9 | 8.1 | -1.0 | - | - | 37.5 | **71.5** | 70.7 $\pm$ 5.2 |
| pen-cloned | 37 | 1.6 | 26.5 | - | - | 39.2 | 37.3 | **58.2** $\pm$ 1.6 |
| hammer-human | 1.2 | 0.3 | 0.3 | - | - | **4.4** | 1.4 | 4.1 $\pm$ 1.2 |
| hammer-cloned | 0.6 | 0.3 | 0.3 | - | - | **2.1** | **2.1** | 2.2 $\pm$ 1.4 |
| door-human | 2 | -0.3 | -0.3 | - | - | **9.9** | 4.3 | 1.5 $\pm$ 0.5 |
| door-cloned | 0.0 | -0.1 | -0.1 | - | - | 0.4 | 1.6 | **2.2** $\pm$ 0.7 |
| **adroit-total** | 104.7 | 9.9 | 25.7 | - | - | 93.5 | 118.2 | **138.9** |
| **kitchen+antmaze+adroit** | 359.4 | - | - | - | - | 541.7 | 656.0 | **827.6** |

**Baselines** We mainly compare our method with two state-of-the-art methods CQL (Kumar et al., 2020) and IQL (Kostrikov et al., 2021b) in all the above-mentioned datasets. Both CQL and IQL are based on Q-learning with constraints on the Q function to alleviate the OOD action issue in the offline setting. In addition, we also report the performance of BRAC-p (Wu et al., 2019), BEAR (Kumar et al., 2019), DT (Chen et al., 2021), and AWAC (Nair et al., 2020) in the datasets they used. Performance of behavior cloning (BC) is from (Kostrikov et al., 2021b).

**Hyperparameters** In all the experiments, the dimension of metric space is set to 10. The margin $m$ in Eq.1 and 2 is 1. The distance threshold $\gamma_m$ is set to 0.5, 0.8, and 0.3 in Kitchen, AntMaze, and Adorit, separately. Hyperparameters are selected from 12 configurations. We use Adam optimizer (Kingma & Ba, 2014) with a learning rate $10^{-3}$, train the model for 800 epochs with batch size 100, and select the best-performing checkpoint. More details about hyperparameters and experiment settings are in Appx.D.

**Performance** Experimental results are shown in Tab.1. VMG's scores are averaged over three individually trained models and over 100 individually evaluated episodes in the environment. In general, VMG outperforms baseline methods in Kitchen and AntMaze and shows competitive performance in Adroit. Note that a good reasoning ability in Kitchen and AntMaze domains is crucial as the rewards in both domains are sparse, and the agent needs to plan over a long time before getting reward signals. In AntMaze, baseline methods perform relatively well in the smallest maze 'umaze', which requires less than 200 steps to solve. In the maze 'large' where episodes can be longer than 600 steps, the performance of baseline methods drops dramatically. VMG keeps a reasonable score in all three mazes, which suggests that simplifying environments to a graph-structured MDP helps RL methods better reason over a long horizon in the original environment. Adroit is the most challenging domain for all the methods in D4RL with a high-dimensional action space. VMG still shows competitive performance in Adroit compared to baselines. Experiments show that learning a policy directly in VMG helps agents perform well, especially in environments with sparse rewards and long temporal horizons.

## 4.2 UNDERSTANDING VALUE MEMORY GRAPH

To analyze whether VMG can understand and represent the structure of the task space correctly, we visualize an environment, the corresponding VMG, and their relationship in Fig.4. We study the task "antmaze-large-diverse" shown in Fig.4a as the state space of navigation tasks is easier to visualize and understand. The target location where the agent receives a positive reward is denoted by a red circle. A successful trajectory is plotted as the green path. To visualize VMG, all the state features $f_s$ are reduced to two dimensions via UMAP (McInnes et al., 2018) and used as the coordinate to plot corresponding vertices as shown in Fig.4b. The graph state values are denoted by color shades. Vertices with darker blue have higher values. As shown in Fig.4b, VMG allocates high values to vertices that are close to the target location and low values to far away vertices. Besides, the topology of VMG is similar to the maze. This is further visualized in Fig.4c where graph vertices are mapped to the corresponding maze locations to show their relationship. Our analysis suggests that VMG can

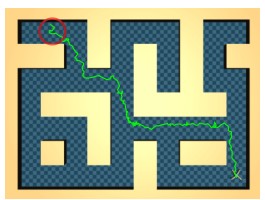
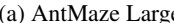
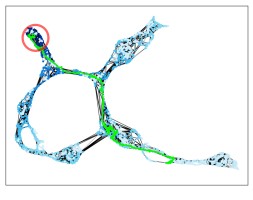
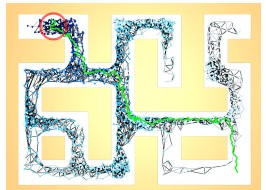

|  (a) AntMaze Large  |  (b) VMG  |  (c) VMG mapped to the maze  |

Figure 4: An example of VMG learned from the dataset 'antmaze-large-diverse'. Fig.4a shows the environment with the target location highlighted by a red circle. VMG is visualized via UMAP in Fig.4b. Graph state values are represented by color shades with higher values in darker blue. Graph states that are close to the target have high values calculated by value iteration. In Fig.4c, graph states are mapped to the corresponding maze locations to show the relationship.

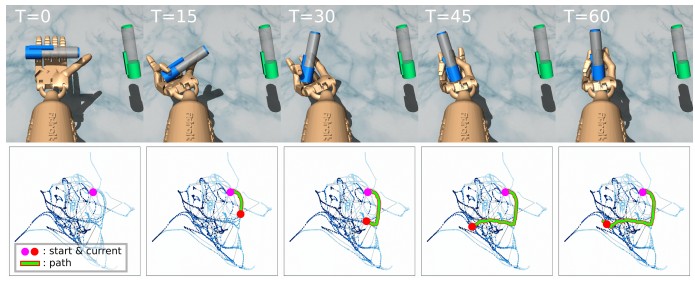

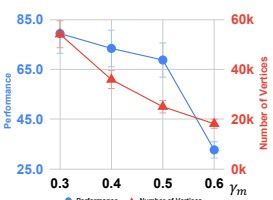

Figure 5: VMG and a successful trial in the task "pen-human". The blue pen is rotated to the same orientation as the green one.

Figure 6: Influence of $\gamma_m$ in "kitchen-partial" in performance and VMG size.

learn a meaningful representation of the task. Another VMG visualization in the more complicated task "pen-human" is shown in Fig.5 and and more visualizations can be found in Appx.J.

### 4.3 REUSABILITY OF VMG WITH NEW REWARD FUNCTIONS

In offline RL, policies are trained to master skills that can maximize accumulated returns via an offline dataset. When the dataset contains other skills that don't lead to high rewards, these skills will be simply ignored. We name them ignored skills. We can retrain a new policy to master ignored skills by redefining new reward functions correspondingly.

Table 2: VMG success rate of ignored skills. Agents can perform these skills by rerunning value iteration with the new reward function in a trained VMG.

| Value Iteration on | Bottom Burner | Top Burner | Hinge Cabinet |
|---|---|---|---|
| Orig. Reward | 0.7 | 3.7 | 0.0 |
| New Reward | **69.7** | **88.3** | **7.3** |

However, rerunning RL methods with new reward functions is cumbersome in the original complex environment, as we need to retrain the policy network and Q/value networks from scratch. In contrast, rerunning value iteration in VMG with new reward functions takes less than one second without retraining any neural networks. Note that the learning of VMG and the action translator is reward-free. Therefore, we don't need to retrain VMG but recalculate graph rewards using Eq.6 with new reward functions.

We design an experiment in the dataset "kitchen-partial" to verify the reusability of VMG with new reward functions. In this dataset, the robot only receives rewards in the following four subtasks: open a microwave, move a kettle, turn on a light, and open a slide cabinet. Besides, there are training episodes containing ignored skills like turning on a burner or opening a hinged cabinet. We first train a model in the original dataset. Then, we define a new reward function, where only ignored skills have positive rewards and relabel training episodes correspondingly. After that, we recalculate graph rewards using Eq.6, rerun value iteration on VMG, and test our agent. Experimental results in Tab.2 show that agents can perform ignored skills after rerunning value iteration in the original VMG with recalculated graph rewards without retraining any neural networks.

### 4.4 ABLATION STUDY

**Distance Threshold**    The distance threshold $\gamma_m$ directly controls the "radius" of vertices and affects the size of the graph. We demonstrate how $\gamma_m$ affects the performance in the task "kitchen-partial" in Fig.6. The dataset size of "kitchen-partial" is 137k. A larger $\gamma_m$ can reduce the number of vertices but hurts the performance due to information loss. More results can be found in Appx.E.

**Graph Reward**    Here we study how will different designs of the graph reward $R_G(v_{j_1}, v_{j_2})$ affect the final performance. In addition to the original version defined in Eq.5 and Eq.6 that averages over the environment rewards, we try maximization and summation and denote them as $R_{G,max}$ and $R_{G,sum}$, separately. Besides, we also study the effectiveness of the the internal reward through the following three variants of Eq.6: $R_{G,rm} = R_{j_1,j_2}$, $R_{G,rm,h} = R_{j_1,j_2} + \frac{1}{2}R_{j_2,j_2}$, $R_{G,rm,t} = \frac{1}{2}R_{j_1,j_1} + R_{j_1,j_2}$, if $e_{j_1 \to j_2} \in \mathcal{E}$. Experimental results shown in Tab.3 suggest that the original design of the graph rewards represent the environment well and leads to the best performance.

**Importance of Contrastive Loss**    The contrastive loss is the key training objective to learning a meaningful metric space. The contrastive loss pushes states that can be reached in a few steps to be close to each other and pushes away other states. To verify this, we train a variant of VMG without the contrastive loss and show the results in Tab.4. The variant without the contrastive

Table 3: Ablation study of graph reward design in VMG. The original design gives us the best performance.

| Variants | kitchen-partial | antmaze-medium-play | pen-human |
|---|---|---|---|
| $R_{G,max}$ | 31.5 | 48.3 | 50.8 |
| $R_{G,sum}$ | 50.3 | 56.3 | 66.4 |
| $R_{G,rm}$ | 55.0 | 55.0 | 65.2 |
| $R_{G,rm,h}$ | 54.6 | 78.0 | **72.2** |
| $R_{G,rm,t}$ | 67.0 | 78.5 | 65.9 |
| $R_G$ (Orig.) | **68.8** | **82.7** | 70.7 |

loss does not work at all (0 scores) in the 'kitchen-partial' and 'antmaze-medium-play' tasks, and the performance in 'pen-human' significantly decreases from 70.7 to 41.2. Results indicate the importance of contrastive loss in learning a robust metric space.

**Effectiveness of Action Decoder**    The action decoder is trained to reconstruct the original action from the transition in the metric space conditioned on the state feature shown in Fig.2b. In this way, the training of the action decoder encourages transitions in the metric space to better represent actions and leads to a better metric space.

Table 4: Ablation study of contrastive loss, action decoder, and Dijkstra search.

| Model | kitchen-partial | antmaze-medium-play | pen-human |
|---|---|---|---|
| VMG | **68.8** | **82.7** | 70.7 |
| - contrastive loss | 0.0 | 0.0 | 41.2 |
| - action decoder | 15.4 | 66.3 | 68.5 |
| - multi-step search | 39.8 | 61.5 | **72.1** |

To show the effectiveness of the action decoder, we train a VMG variant without the action decoder and show the results in Tab.4. The performance without the action decoder drops in all three tested tasks, especially in 'kitchen-partial' (from 68.8 to 15.4). The results verify our design choice.

**Multi-Step Search**    In Tab.4, we list the performance of our method without Multiple-Step Search. Compared to the original version, we observe a performance drop in 'kitchen-partial' and 'antmaze-medium-play' and similar performance in 'pen-human', which suggests that instead of greedily searching one step in the value interaction results, searching multiple steps first to find a high value state in the long future and then plan a path to it via Dijkstra can help agents perform better. We think the advantage might caused by the gap between VMG and the environment. An optimal path on VMG searched directly by value iteration may not be still optimal in the environment. At the same time, a shorter path from Dijkstra helps reduce cumulative errors and uncertainty, and thus increases the reliability of the policy.

**Limitations**    As an attempt to apply graph-structured world models in offline reinforcement learning, VMG still has some limitations. For example, VMG doesn't learn to generate new edges in the graph but only creates edges from existing transitions in the dataset. This might be a limitation when there is not enough data provided. In addition, VMG is designed in an offline setting. Moving to the online setting requires further designs for environment exploring and dynamic graph expansion, which can be interesting future work. Besides, the action translator is trained via conditioned behavior cloning. This may lead to suboptimal results in tasks with important low-level dynamics like gym locomotion (See Appx.H). Training the action translator by offline RL methods may alleviate this issue.

## 5    CONCLUSION

We present **V**alue **M**emory **G**raph (VMG), a graph-structured world model in offline reinforcement learning. VMG is a Markov decision process defined on a directed graph trained from the offline dataset as an abstract version of the environment. As VMG is a smaller and discrete substitute for the original environment, RL methods like value iteration can be applied on VMG instead of the original environment to lower the difficulty of policy learning. Experiments show that VMG can outperform baselines in many goal-oriented tasks, especially when the environments have sparse rewards and long temporal horizons in the widely used offline RL benchmark D4RL. We believe VMG shows a promising direction to improve RL performance via abstracting the original environment and hope it can encourage more future works.

ACKNOWLEDGMENTS

We would like to thank Ahmed Hefny and Vaneet Aggarwal for their helpful feedback and discussions on this work.

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

CONTENTS

## A    ENVIRONMENT DETAILS

The datasets in D4RL (Fu et al., 2020) is under CC BY license and the related code is under Apache 2.0 License. We use the latest version of the datasets (v1/v0/v1 for AntMaze, Kithcen, Adroit, separately). Different versions of datasets contain exactly the same training transitions. The newer version fixes some bugs in the meta data information like the wrong termination steps. Performance is measured by returns normalized to the range between 0 and 100 defined by the D4RL benchmark [9]. In detail, $normalized\ score = 100 \times \frac{score - random\ score}{expert\ score - random\ score}$. A score of 100 corresponds to the average returns of a domain-specific expert. For AntMaze, and Kitchen, an estimate of the maximum score possible is used as the expert score. For Adroit, this is estimated from a policy trained with behavioral cloning on human-demonstrations and online fine-tuned with RL in the environment. For more details about the datasets please refer to D4RL (Fu et al., 2020).

## B    ALGORITHMS

### B.1    GRAPH CONSTRUCTION

The detailed algorithm of graph construction is shown in Alg.1.

---

**Algorithm 1:** Graph Construction

**Input :** Training Set $\mathcal{D} = \{(s_i, a_i, r_i, s_i') | i = 1, 2, ..., N\}$, Empty vertices set $\mathcal{V} = \{\}$,
     Current vertex index $J = 1$, Distance threshold $\gamma_m$, Empty edges set $\mathcal{E} = \{\}$

1 **for** $(s_i, a_i, r_i, s_i')$ *in* $\mathcal{D}$ **do**
2  $f_{s_i} = Enc_s(s_i)$
3  Compute the distance $d_{ij}$ between $f_{s_i}$ and $f_{v_j}$ for every $f_{v_j}$ in $\mathcal{V}$
4  **if** $\min\{d_{ij} | f_{s_j}\ in\ \mathcal{V}\} > \gamma_m\ or\ J = 1$ **then**
5   $v_J \leftarrow s_i, f_{v_J} \leftarrow f_{s_i}$
6   $\mathcal{V}$.append($(v_J, f_{v_J})$)
7   $J \leftarrow J + 1$
8  **end**
9 **end**
10 **for** $(s_i, a_i, r_i, s_i')$ *in* $\mathcal{D}$ **do**
11  Find $v_{j_1}, v_{j_2}$ that $s_i$ and $s_i'$ are classified to in $\mathcal{V}$, respectively
12  **if** $v_{j_1} \neq v_{j_2}$ *and* the connection $e_{j_1 \rightarrow j_2} \notin \mathcal{E}$ **then**
13   $\mathcal{E}$.append($e_{j_1 \rightarrow j_2}$)
14  **end**
15 **end**

---

### B.2    POLICY EXECUTION

The detailed algorithm of policy execution is shown in Alg.2.

**Details of Dijkstra**  When we use Dijkstra in Sec.3.4 to plan a path $\mathcal{P}$ from $v_c$ to $v^*$, we define weights to each edge to make sure $\mathcal{P}$ is both short and high-rewarded. The weights used to plan the path $\mathcal{P}$ are based on rewards. For each edge $e_{j_1 \rightarrow j_2}$, we define the edge weight $w_{j_1 \rightarrow j_2}$ as the gap between the maximal graph reward and the edge reward and denote the weight set as $\mathcal{W}$. $w_{j_1 \rightarrow j_2} = \max\{R_G(v_{j_3}, v_{j_4}) | \forall e_{j_3 \rightarrow j_4} \in \mathcal{E}\} - R_G(v_{j_1}, v_{j_2})$.

## C    ARCHITECTURE OF NEURAL NETWORKS

For all the networks including the state encoder $Enc_s$, the action encoder $Enc_a$, the action decoder $Dec_a$, and the action translator $Tran(s, s')$, we use a 3-layer MLP with hidden size 256 and ReLU activation functions.

---

**Algorithm 2:** Policy Execution

---

**Input :** Current state $s_c$, State encoder $Enc_s$, Action translator $Tran$, Vertex and edge sets in
VMG $(\mathcal{V}, \mathcal{E})$, Vertices value $V$, Edge weight $\mathcal{W}$

**1** $f_{s_c} = Enc_s(s_c)$

**2** $v_c = \arg\min_{v_j|(v_j,f_j)\in\mathcal{V}} D(f_{s_c}, f_j)$

**3** Search future horizon of $N_s$ steps starting from $v_c$ and select the best value vertex $v^*$

**4** Compute the weighted shortest path $\mathcal{P}$ from $v_c$ to $v^*$ via Dijkstra. $\mathcal{P} = [v_c, v_{c+1}, ..., v^*]$

**5** $a_c = Tran(s_c, v_{c+N_{sg}})$

---

## D    EXPERIMENT SETTINGS AND HYPERPARAMETERS

Our model is trained in a single RTX Titan GPU in about 1.5 hours. For inference, building the graph from clustering takes about 0.5-2 minutes before the evaluation. After that, it takes about 0.5-10 minutes to evaluate 100 episodes. We implement VMG on the top of the offline RL python package d3rlpy (Takuma Seno, 2021) with MIT license. In all the experiments, We use Adam optimizer (Kingma & Ba, 2014) with a learning rate $10^{-3}$. Batch size is 100. Each model is trained for 800 epochs. We save models per 50 epochs and report the performance of the best one evaluated in the environment from the checkpoints saved from the 500th to the 800th epochs. The remaining hyperparameter settings can be found in Tab.5. $N_s = \infty$ means we search the future steps till the end of the graph. For the domain Kitchen, the hyperparameters are tuned in "kitchen-partial". For AntMaze it is "antmaze-umaze-diverse". For Adroit, hyperparameters are tuned individually. We use the environment to tune the hyperparameters. We searched four hyperparameters in our main experiments: $\gamma_m$ in [0.3, 0.5, 0.8, 1.0, 1.2], reward discount in [0.8, 0.95], $N_{sg}$ in [1, 2, 3], $N_s$ in [12, $\infty$]. Hyperparameters are searched one by one, in total 12 configurations. For hyperparameters like batch size or learning rate, we follow the default one in the RL library d3rlpy. The dimension of the metric space is set to 10 in all the experiments. Tuning the hyperparameters offline is an ongoing and important research topic in offline RL, and we left it for future work.

Table 5: Detailed Hyperparameter Setting

| Dataset | $m$ | $K$ | $\gamma_m$ | discount | $N_{sg}$ | $N_s$ |
|---|---|---|---|---|---|---|
| kitchen-complete | 1 | 10 | 0.5 | 0.95 | 2 | $\infty$ |
| kitchen-partial | 1 | 10 | 0.5 | 0.95 | 2 | $\infty$ |
| kitchen-mixed | 1 | 10 | 0.5 | 0.95 | 2 | $\infty$ |
| antmaze-umaze | 1 | 10 | 0.8 | 0.8 | 1 | $\infty$ |
| antmaze-umaze-diverse | 1 | 10 | 0.8 | 0.8 | 1 | $\infty$ |
| antmaze-medium-play | 1 | 10 | 0.8 | 0.8 | 1 | $\infty$ |
| antmaze-medium-diverse | 1 | 10 | 0.8 | 0.8 | 1 | $\infty$ |
| antmaze-large-play | 1 | 10 | 0.8 | 0.8 | 1 | $\infty$ |
| antmaze-large-diverse | 1 | 10 | 0.8 | 0.8 | 1 | $\infty$ |
| pen-human | 1 | 10 | 0.3 | 0.8 | 2 | 12 |
| pen-cloned | 1 | 10 | 0.3 | 0.8 | 2 | 12 |
| hammer-human | 1 | 10 | 1.0 | 0.8 | 2 | 12 |
| hammer-cloned | 1 | 10 | 1.0 | 0.8 | 2 | 12 |
| door-human | 1 | 10 | 0.3 | 0.8 | 2 | 12 |
| door-cloned | 1 | 10 | 0.3 | 0.8 | 2 | 12 |

## E    ABLATION STUDIES

### E.1    DISTANCE THRESHOLD

More experimental results of the distance threshold $\gamma_m$ in the tasks "antmaze-medium-play" and "pen-cloned" can be found in Fig.7. Results suggest that the model is not so sensitive to $\gamma_m$ if it is not too large.

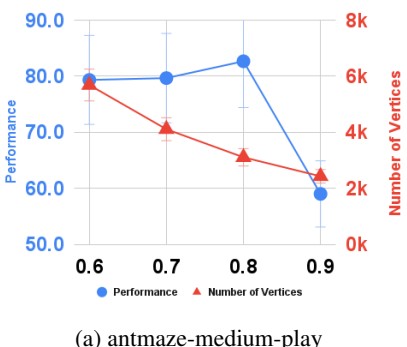
(a) antmaze-medium-play

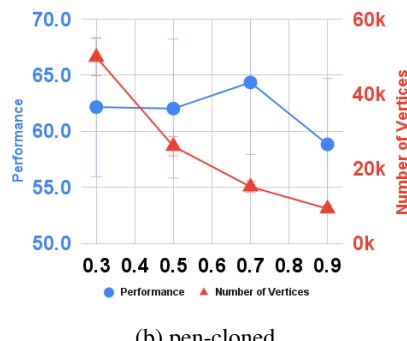
(b) pen-cloned

Figure 7: More results of the influence of $\gamma_m$ in performance and VMG size

## E.2 STATE MERGING METHOD

Vertices in VMG are merged from the original states based on a distance threshold $\gamma_m$ as described in Sec.3.2. It is also possible to use other clustering methods to merge states. However, the dataset sizes in some tasks can be up to 1 million. Many advanced clustering methods (like BIRCH (Zhang et al., 1996)) are slow in this case (up to hours for BIRCH on machines with Intel Xeon Gold 6242). Therefore, we compare with the classical K-means (Lloyd, 1982) implemented on Faiss (Johnson et al., 2019) library with the GPU support in both the AntMaze domain and the Kitchen domains. Faiss-based K-means can be finished up to 20 seconds in our setting. Our merging method takes up to 1 minute. Experimental results are shown in Tab.6. VMG created by our merging method performs better than the one created by K-means. The vertices of our method can be viewed as hyperspheres in the metric space with the same radius $\gamma_m$. In contrast, K-means cannot directly specify the size of each cluster, which can result in vertices with different "volumes" in the metric space. This might lead to undesired distortion in the graph and reduce the performance. As K-means doesn't have a parameter to control the size of the clusters directly, we have to search for the best number of clusters for every dataset. The number of clusters used in K-means is shown in Tab.7.

Table 6: Ablation study of different state merging methods. Our original design gives us better performance.

| Model | AntMaze | | | | | | Kitchen | | |
| | umaze | umaze-diverse | medium-play | medium-diverse | large-play | large-diverse | complete | partial | mixed |
| --- | --- | --- | --- | --- | --- | --- | --- | --- | --- |
| VMG with K-means | 88.7 | 79.7 | 81.2 | 77.0 | **72.3** | **76.3** | 61.1 | 18.3 | 43.6 |
| VMG | **93.7** | **94.0** | **82.7** | **84.3** | 67.3 | 74.3 | **73.0** | **68.8** | **50.6** |

Table 7: Number of clusters used in K-means

| AntMaze | | | | | | Kitchen | | |
| umaze | umaze-diverse | medium-play | medium-diverse | large-play | large-diverse | complete | partial | mixed |
| --- | --- | --- | --- | --- | --- | --- | --- | --- |
| 6000 | 2000 | 1000 | 1000 | 10000 | 10000 | 3000 | 25000 | 25000 |

## E.3 INFLUENCE OF $m$

The value of the margin $m$ in Eq.1 and Eq.2 implicitly defines the minimal distance of negative state pairs in the learned metric space. To study the influence of $m$ on the performance, here we set m to 0.5, 1, and 2 in the datasets antmaze-medium-play, kitchen-partial, and pen-human and show the results in Tab.8. In the antmaze experiment, performance becomes better with a larger m. But in the task pen-human, a smaller m gives us better results. In kitchen-partial, m=1 shows the best performance. Experimental results suggest that m=1 is a reasonable value for the tasks we evaluate on. And if we tune m separately, it is possible to improve the performance.

| m | 0.5 | 1 | 2 |
|---|---|---|---|
| antmaze-medium-play | 65.0 | 82.7 | 84.0 |
| kitchen-partial | 17.0 | 64.5 | 68.8 |
| pen-human | 75.1 | 70.7 | 65.7 |

Table 8: Influence of the margin $m$.

### E.4    INFLUENCE OF DISCOUNT FACTOR

Here we study how different discount factor values will affect the performance of VMG. We set the discount factor to the values 0.8, 0.95, and 0.99. Experiments in Tab.9 show that 0.99 leads to better performance in pen-human and comparable performance in kitchen-partial. In antmaze-medium-play, 0.99 performs worse than 0.8 and 0.95, which suggest that a small discount factor in antmaze might help reduce cumulative errors.

| discount factor | 0.8 | 0.95 | 0.99 |
|---|---|---|---|
| antmaze-medium-play | 82.7 | 76.3 | 75.0 |
| kitchen-partial | 58.2 | 68.8 | 68.1 |
| pen-human | 70.7 | 69.0 | 74.8 |

Table 9: Influence of the discount factor.

### E.5    INFLUENCE OF THE DIMENSION OF THE METRIC SPACE

Here we study how different numbers of the metric space dimension will affect the performance of VMG. We set the metric space dimensions to 5, 10, and 20. Experiments in Tab.10 show that models with latent space dimensions 10 and 20 perform better than those with 5, which suggests that a reasonable performance requires big enough dimensions of the latent space to represent the states and actions better. Besides, space with 10 dimensions works better than 20 in kithcen-partial but worse than 20 in pen-human, this suggests the performance has space to improve if we tune the dimensions individually in each task.

| metric space dim | 5 | 10 | 20 |
|---|---|---|---|
| antmaze-medium-play | 71.0 | 82.7 | 82.0 |
| kitchen-partial | 5.75 | 68.8 | 46.0 |
| pen-human | 70.7 | 70.7 | 79.0 |

Table 10: Influence of the metric space dimension.

### E.6    INFLUENCE OF $K$

The hyperparameter $K$ used in training the action translator in Sec.3.4 defines the range of the future states the action translator conditions on during training. To study the influence of $K$, here show experiments with $K$=5, 10, and 20 in Tab.11. We notice that $K = 5$ doesn't work in antmaze-medium and kitchen-partial, which suggests that K=5 is not big enough to cover 2 steps in the graph transition. In addition, the experiments with $K = 20$ show better results than $K = 10$ in kitchen-partial and pen-human. In antmaze-medium-play, $K = 10$ performs the best. Experimental results suggest that a big enough $K$ helps the model perform better.

## F    TRAINING CURVE

Fig.8 shows the training curves of the contrastive loss $L_c$, the action loss $L_a$, and the anction translator loss $L_{Tran}$ in tasks kitchen-partial, antmaze-medium-play, and pen-human.

| $K$ | 5 | 10 | 20 |
|---|---|---|---|
| antmaze-medium-play | 7.0 | 82.7 | 74.0 |
| kitchen-partial | 0.3 | 68.8 | 76.8 |
| pen-human | 80.3 | 70.7 | 83.5 |

Table 11: Influence of $K$.

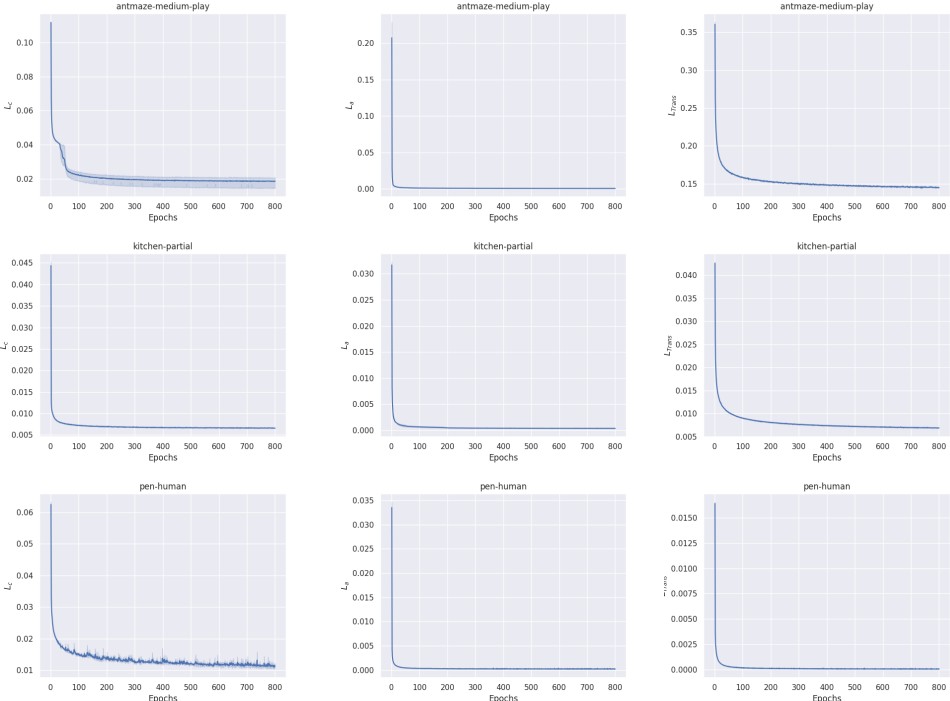

Figure 8: Training curve of the contrastive loss $L_c$, action loss $L_a$, and action translator loss $L_{Tran}$.

## G  ENVIRONMENT/GRAPH TRANSITION RATIO

VMG abstracts the original continuous environment into a finite and relatively small graph. An one-step transition in the graph corresponds to multiple steps in the environment. Here we compute the average numbers of environment transitions per graph transition in our main experiments and list the results in Tab.12.

Table 12: Average number of environment transitions per graph transition.

| | AntMaze | | | | | Kitchen | | | Adroit | | | | | |
|---|---|---|---|---|---|---|---|---|---|---|---|---|---|---|
| umaze | | medium | | large | | | | | pen | | hammer | | door | |
| - | diverse | play | diverse | play | diverse | complete | partial | mixed | human | cloned | human | cloned | human | cloned |
| 4.4 | 3.3 | 14.4 | 15.1 | 10.3 | 9.7 | 1.1 | 2.0 | 2.0 | 1.8 | 4.9 | 3.1 | 9.9 | 1.0 | 1.5 |

## H  EXPERIMENTS IN GYM LOCOMOTION TASKS

VMG is introduced to help agents reason the long future better so as to improve their performance in complex environments with sparse rewards and large search space due to long temporal horizons and continuous state/action spaces. VMG may not help in gym locomotion tasks, as these tasks don't require agents to reason the long future and thus are out of our scope. Gym locomotion tasks provide rich and dense reward signals, and the motion patterns to learn in these tasks are periodic and short. Therefore, the problems VMG designed to solve are not an issue here. Our performance in these tasks

is expected to be close to behavior cloning, since the low-level component, the action translator, is trained via (conditioned) behavior cloning. The action translator is used to handle local dynamics that are not modeled in VMG. Here we run new experiments in these tasks and show the results below. Experimental results verify our assumption. Results and analysis suggest that an improved design and/or learning strategy of the action translator might help improve the performance. For example, training the action translator using conditioned offline RL methods instead of conditioned behavior cloning. However, this is orthogonal to our VMG framework contribution to future reasoning, and we leave it for future work.

| Dataset | VMG | BC | CQL | IQL |
|---|---|---|---|---|
| halfcheetah-medium | 42.2 | 42.6 | 44.0 | 47.4 |
| hopper-medium | 49.4 | 52.9 | 58.5 | 66.3 |
| walker2d-medium | 70.4 | 75.3 | 72.5 | 78.3 |
| halfcheetah-medium-replay | 38.2 | 36.6 | 45.5 | 44.2 |
| hopper-medium-replay | 15.2 | 18.1 | 95.0 | 94.7 |
| walker2d-medium-replay | 28.6 | 26.0 | 77.2 | 73.9 |
| halfcheetah-medium-expert | 80.5 | 55.2 | 91.6 | 86.7 |
| hopper-medium-expert | 49.5 | 52.5 | 105.4 | 91.5 |
| walker2d-medium-expert | 70.4 | 107.5 | 108.8 | 109.6 |

Table 13: Peformance of VMG in gym locomotion tasks. The performance of VMG is expected to be closed to behavior cloning.

## I  FUTURE WORKS

There are several directions to improve VMG. Building hierarchical graphs to model different levels of environment structures might help represent the environment better. For example, if a robot needs to cook a meal, we might have a high-level graph to represent abstract tasks like washing vegetables, cutting vegetables, etc. A low-level graph can be used to guide a goal-conditioned policy. This might improve the high-level planning of the tasks. Extending VMG into the online setting is also an important future step. In online reinforcement learning, data with new information is collected throughout the training stage. Therefore, the graph needs to have a mechanism to continually expand and include the new information. Besides, exploration is a crucial component in online reinforcement learning. If we model the uncertainty of the graph, VMG can be used to guide the agent to explore regions with high uncertainty to explore more effectively. Combined with Monte Carlo tree search on VMG might also help policy explore and exploit better.

## J  VISUALIZATION OF VMG

More visualization of VMG in different tasks are demonstrated in Fig.9, 10, 11, 12. An episode is denoted as a green path on the graph with a "+" sign at the end.

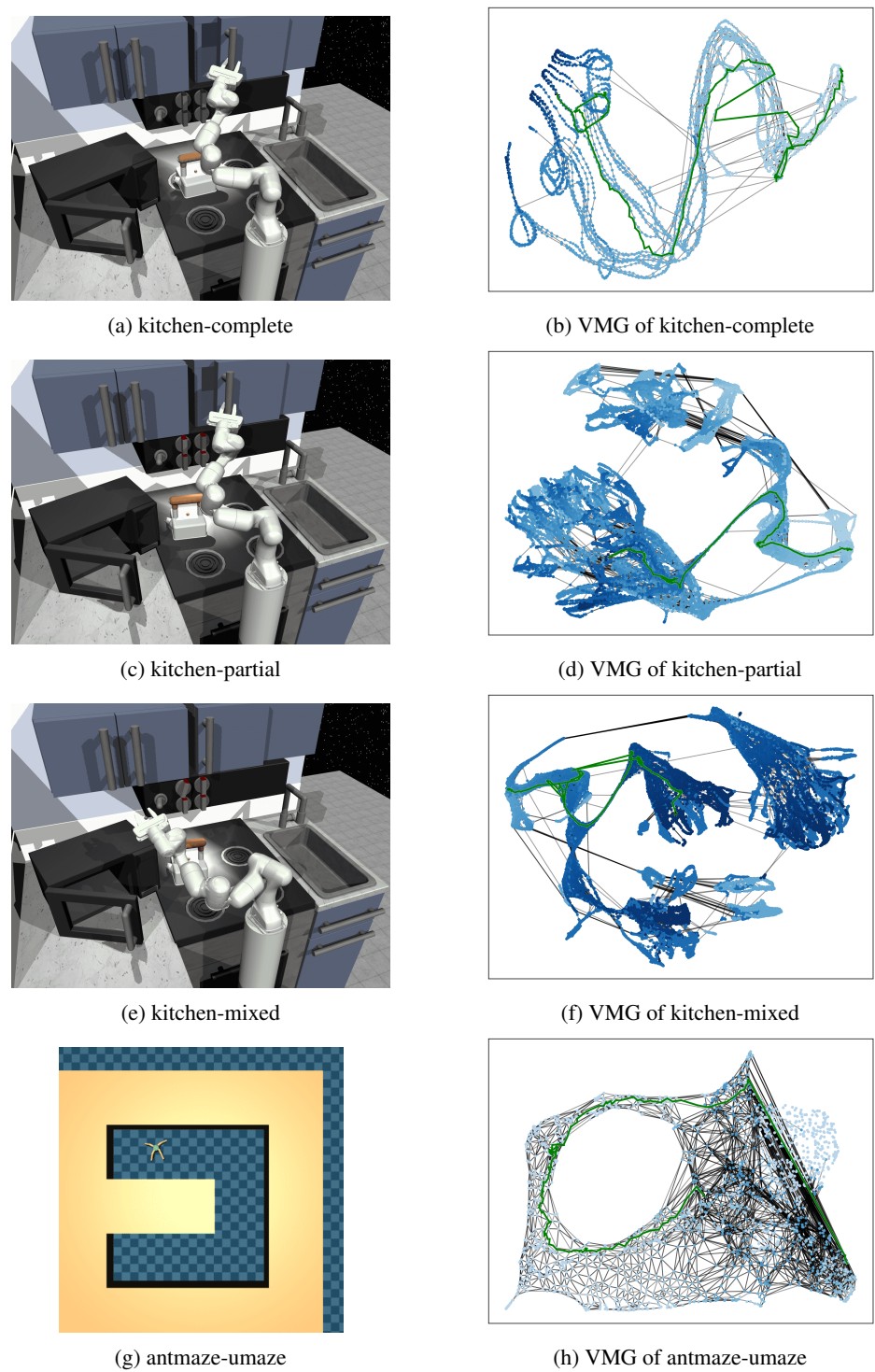

(a) kitchen-complete

(b) VMG of kitchen-complete

(c) kitchen-partial

(d) VMG of kitchen-partial

(e) kitchen-mixed

(f) VMG of kitchen-mixed

(g) antmaze-umaze

(h) VMG of antmaze-umaze

Figure 9: Visualization of VMG in different tasks

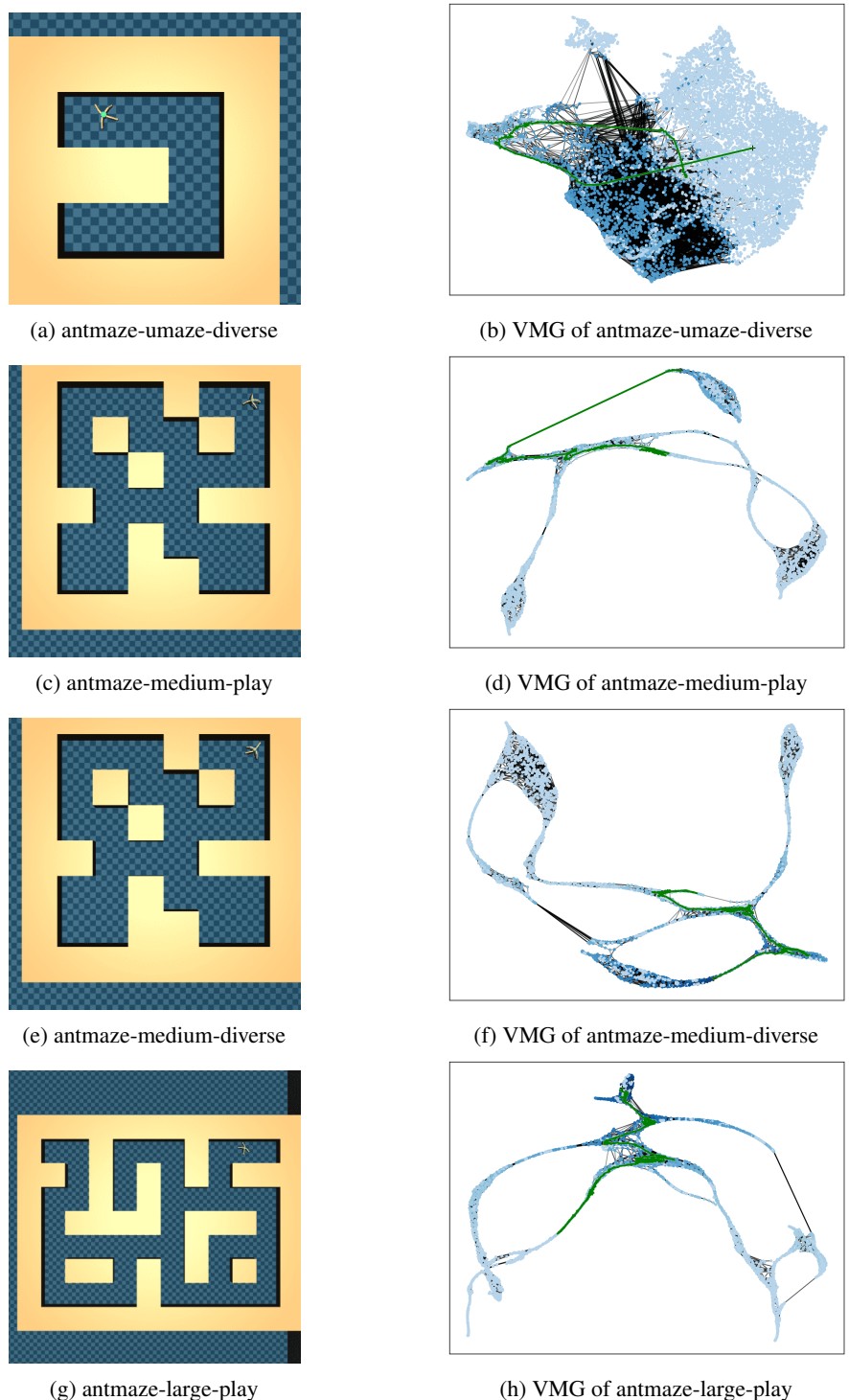

(a) antmaze-umaze-diverse

(b) VMG of antmaze-umaze-diverse

(c) antmaze-medium-play

(d) VMG of antmaze-medium-play

(e) antmaze-medium-diverse

(f) VMG of antmaze-medium-diverse

(g) antmaze-large-play

(h) VMG of antmaze-large-play

Figure 10: Visualization of VMG in different tasks

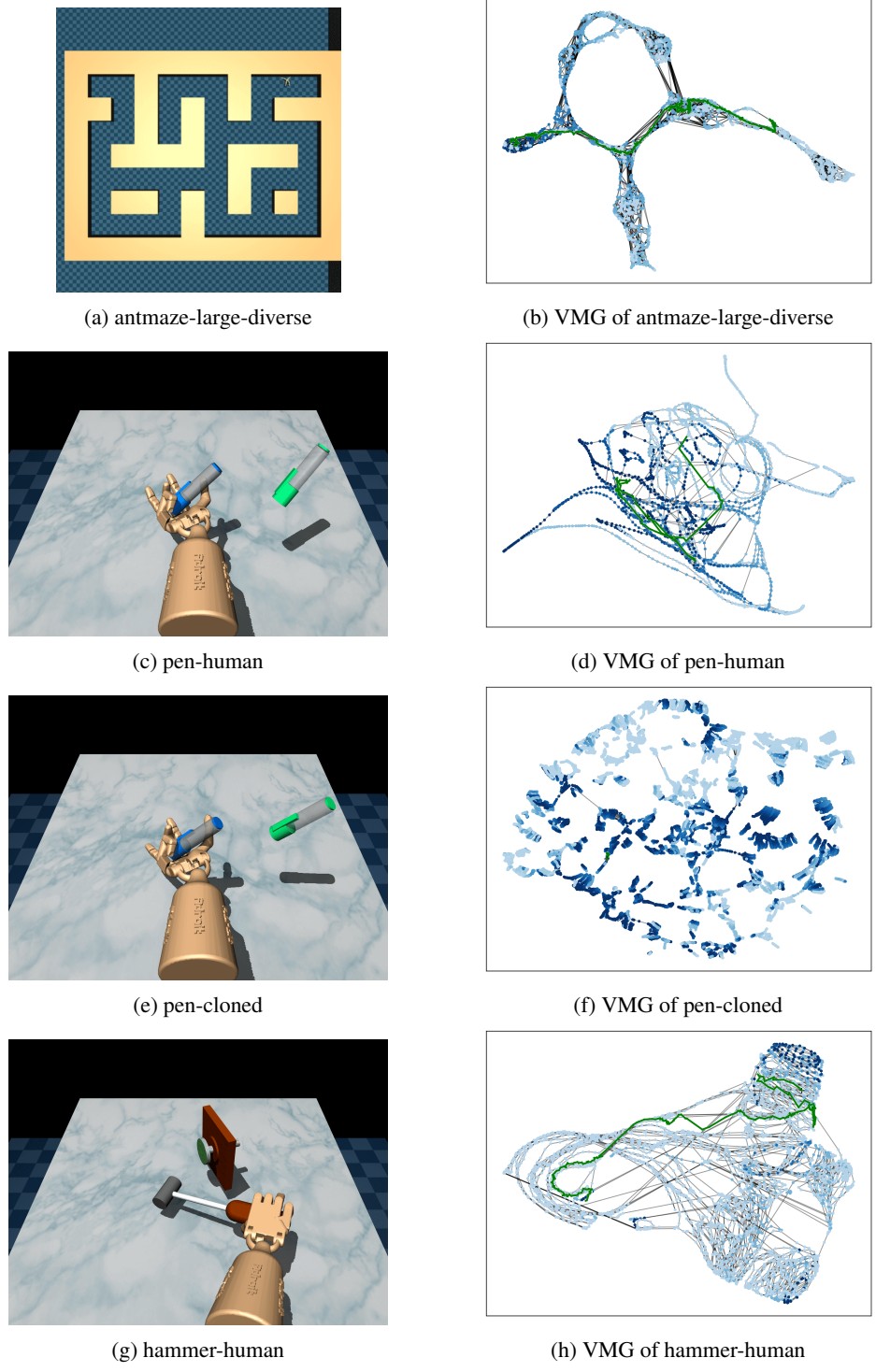

Figure 11: Visualization of VMG in different tasks

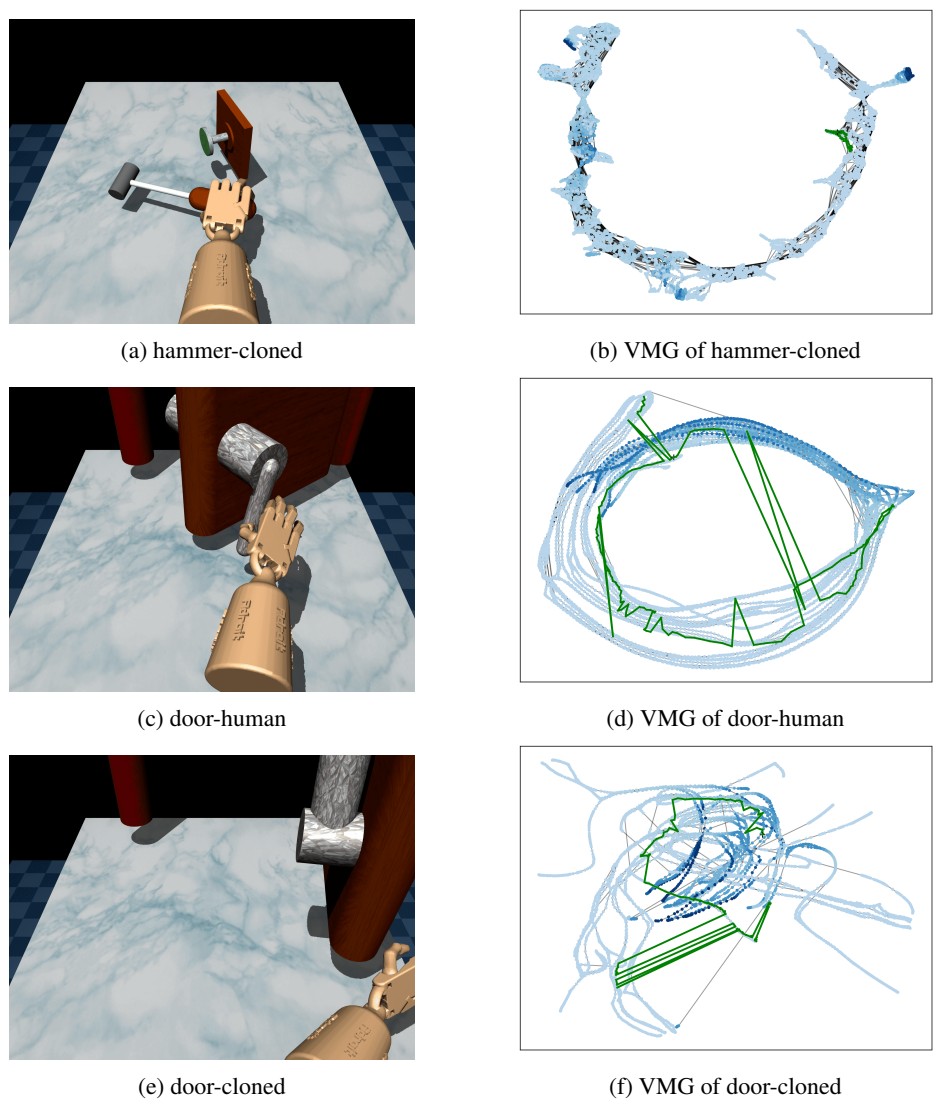

(a) hammer-cloned

(b) VMG of hammer-cloned

(c) door-human

(d) VMG of door-human

(e) door-cloned

(f) VMG of door-cloned

Figure 12: Visualization of VMG in different tasks

