# OpenReview forum: "Value Memory Graph: A Graph-Structured World Model for Offline Reinforcement Learning"
_ICLR.cc/2023/Conference — ICLR 2023 poster_

### Official Review · Reviewer_b4KN · 2022-10-18

**Confidence:** 4
**Correctness:** 2
**Technical Novelty And Significance:** 3
**Empirical Novelty And Significance:** 3
**Recommendation:** 6

**Clarity, Quality, Novelty And Reproducibility:**

* The writing is clear, and the visualization is of high quality.
* The method is novel, particularly, the approach that organises continuous control data into a graph and the way to get a policy out of the graph where similar states are grouped into a single node.
* For reproducibility, the code and instructions are provided. It seems like one can easily try to reproduce but I didn't try by myself.

**Strength And Weaknesses:**

Strength
* VMG provides a rather new approach to address OOD action problems in offline RL, which may have further impacts.
* The proposed clustering method allows graph-based credit assignment to continuous control.

Weakness
* False claim of SOTA in the abstract. There are lots of offline RL methods that perform better than CQL in d4rl. For model-free methods, there are ATAC[1] and SAC-N[2], etc. And for model-based, there is a Trajectory Transformer[3] and COMBO[4].
* Credit assignment in the transition data graph has been investigated in discrete state/action space in the online RL setting [5,6]. While being able to work in a continuous control is impressive, this work is not the first one that tried to leverage the graph-structured replay data.

Minor problems and questions:
* Why gym-locomotion control results are not presented? I agree the tasks themselves are less interesting but they are most of the offline RL methods are developed and tested on. The empirical results of this work are done on goal-reaching tasks (although some of them, like adroit, have shaped rewards), which might be helpful for the VMG to gain advantages. I would encourage authors to show gym-locomotion results, say, in the appendix if they do have the results. If the VMG indeed struggle with gym-locomotion, it will be better to state the scope of the current state of method clearly.
* I'm not sure if it's appropriate to call VMG a world model. Although the term world model itself is very vague, it usually means a dynamics model plus a reward model. A dynamics model should be a parameterised model that can predict the future state given the current state and action. On the other hand, VMG only memorises and organises existing data and uses it for value estimation and building a policy at test time. To me, this is more like a model-free method with data set organised as a graph.
* How's the training and inference cost of VMG?

Links:
- [1] ATAC ([https://arxiv.org/abs/2202.02446](https://arxiv.org/abs/2202.02446))
- [2] SAC-N ([https://arxiv.org/pdf/2110.01548.pdf](https://arxiv.org/pdf/2110.01548.pdf))
- [3] Trajectory Transformer (https://arxiv.org/abs/2106.02039)
- [4] COMBO [https://arxiv.org/pdf/2102.08363.pdf](https://arxiv.org/pdf/2102.08363.pdf)
- [5] Topological Replay buffer [https://openreview.net/forum?id=OXRZeMmOI7a]
- [6] Graph Backup [https://openreview.net/forum?id=0UQqmPGuL4n]

**Summary Of The Paper:**

The paper proposes to group states in continuous space into clusters, according to a learned reachability metric. Such an operation allows turning the independent transition trajectories into a graph, allowing running value iteration for robust credit assignment. With shortest-path planning and an action translator, the agent can act to maximize value in a test environment while preventing OOD actions.
The empirical evaluations in the d4rl setting show the proposed VMG method surpasses some offline RL methods that applies conventional DQN-style bootstrapping.

**Summary Of The Review:**

I think this is a good paper with significant contributions on the methodology level and the empirical performance of the method is also nice.
Whereas it also has some problems like a false claim of SOTA and missed related work (thus overclaiming contributions).
Also, I have some questions mentioned in Strength And Weaknesses, which might be turned into weaknesses after the author's clarification.

In general, the current version of the paper is not perfect but I'm leaning to accept the paper because the problems should be easily fixed and other potential weaknesses won't surpass the main contributions.

---

> ### Author Response · Authors · 2022-11-15
> **Reply to Reviewer b4KN Part 1**
>
> Thank you for your detailed review, questions, and suggestions. We here address them and will incorporate all the feedback.
>
> **[Q1] False claim of SOTA in the abstract. There are lots of offline RL methods that perform better than CQL in d4rl. For model-free methods, there are ATAC[1] and SAC-N[2], etc. And for model-based, there is a Trajectory Transformer[3] and COMBO[4].**
>
> The baseline methods we mainly compared with, CQL and IQL, provide official performance in the environments we focus on like AntMaze and FrankaKitchen.
> ATAC, SAC-N, and COMBO are not evaluated in the tasks like AntMaze and FrankaKitchen.
> As such environments contain only sparse reward signals with long temporal horizons and are quite different from the benchmark they use like Mujoco gym tasks,
> The performance of ATAC, SAC-N, and COMBO performance in AntMaze and FrankaKitchen is unclear.
> Here we try to apply the codes of ATAC and COMBO AntMaze and FrankaKitchen for a comparison.
> For ATAC, we use the officially released repository [https://github.com/microsoft/ATAC.git].
> As we don't find the official repository of COMBO, we use the COMBO implementation in d3rlpy [https://github.com/takuseno/d3rlpy.git], a widely used offline RL library.
> We use the default hyperparameters they provided in the code due to the limitation of time. Tuning hyperparameters carefully may obtain better performance for ATAC and COMBO. We have contacted the authors for suggestions on hyperparameters.
> The performance we obtained for ATAC and COMBO is worse than CQL, IQL, and VMG in general. Especially in antmaze-medium and antmaze-large, the performance of ATAC and COMBO is close to zero. In contrast, VMG shows relatively well performance due to improved reasoning ability.
>
> Trajectory Transformer (TT) shows a better performance in AntMaze in the small and medium size mazes.
> However, the inference cost of TT makes it impractical to apply in real life.
> Evaluating over 25 episodes takes at least hours in TT due to the huge cost of the beam search with the GPT model.
> The inference cost of our methods is about 1-10 minutes for 100 episodes, depending on the size of the graph and the number of steps we search on the graph.
> Besides, VMG has a better performance in the largest maze, which suggests a better long-horizon reasoning ability.
> In addition, the methods proposed in TT, like using GPT architecture and discretizing state and action space, are orthogonal to our contribution.
> We think combing TT with VMG will be interesting for future work.
>
> | Dataset                | VMG  | TT   | ATAC (produced from official repo) | COMBO (produced by d3rlpy) | CQL  | IQL  |
> |------------------------|------|------|------------------------------------|----------------------------|------|------|
> | kitchen-complete       | 73.0 | -    | 55.0                               | 0.8                        | 43.8 | 62.5 |
> | kitchen-partial        | 68.8 | -    | 35.0                               | 24.5                       | 49.8 | 46.3 |
> | kitchen-mixed          | 50.6 | -    | 32.5                               | 25.0                       | 51.0 | 51.0 |
> | antmaze-umaze          | 93.7 | 100  | 76                                 | 72                         | 74.0 | 87.5 |
> | antmaze-umaze-diverse  | 94.0 | -    | 87                                 | 0                          | 84.0 | 62.2 |
> | antmaze-medium-play    | 82.7 | 93.3 | 0                                  | 0                          | 61.2 | 71.2 |
> | antmaze-medium-diverse | 84.3 | 100  | 1                                  | 1                          | 53.7 | 70.0 |
> | antmaze-large-play     | 67.3 | 66.7 | 0                                  | 0                          | 15.8 | 39.6 |
> | antmaze-large-diverse  | 74.3 | 60.0 | 0                                  | 0                          | 14.9 | 47.5 |

---

> ### Author Response · Authors · 2022-11-15
> **Reply to Reviewer b4KN Part 2**
>
> **[Q2] Why gym-locomotion control results are not presented? If the VMG indeed struggles with gym locomotion, it will be better to state the scope of the current state of method clearly.**
>
> - **VMG is introduced to help agents reason the long future better** so as to improve their performance in complex environments with sparse rewards and large search space due to long temporal horizons and continuous state/action spaces.
>
> - **VMG may not help in gym locomotion tasks, as these tasks don't require agents to reason the long future and thus are out of our scope.** Gym locomotion tasks provide rich and dense reward signals, and the motion patterns to learn in these tasks are periodic and short. Therefore, we think the problems VMG designed to solve are not an issue here.
>
> - **Our performance in these tasks is expected to be close to behavior cloning,** since the low-level component, the action translator, is trained via (conditioned) behavior cloning. The action translator is used to handle local dynamics that are not modeled in VMG. Here we run new experiments in these tasks and show the results below. Experimental results verify our assumption.
>
> - Results and analysis suggest that **an improved design and/or learning strategy of the action translator might help improve the performance.** For example, training the action translator using conditioned offline RL methods instead of conditioned behavior cloning. However, **this is orthogonal to our VMG framework contribution to future reasoning,** and we leave it for future work.
>
> | Dataset                   | VMG  | BC    | CQL   | IQL   |
> |---------------------------|------|-------|-------|-------|
> | halfcheetah-medium        | 42.2 | 42.6  | 44.0  | 47.4  |
> | hopper-medium             | 49.4 | 52.9  | 58.5  | 66.3  |
> | walker2d-medium           | 70.4 | 75.3  | 72.5  | 78.3  |
> | halfcheetah-medium-replay | 38.2 | 36.6  | 45.5  | 44.2  |
> | hopper-medium-replay      | 15.2 | 18.1  | 95.0  | 94.7  |
> | walker2d-medium-replay    | 28.6 | 26.0  | 77.2  | 73.9  |
> | halfcheetah-medium-expert | 80.5 | 55.2  | 91.6  | 86.7  |
> | hopper-medium-expert      | 49.5 | 52.5  | 105.4 | 91.5  |
> | walker2d-medium-expert    | 70.4 | 107.5 | 108.8 | 109.6 |
>
>
>
> **[Q3] I'm not sure if it's appropriate to call VMG a world model. World model usually means a dynamics model plus a reward model.**
>
> VMG is an MDP defined on a graph to abstract the original environment.
> VMG does have a dynamics model (Eq.4) and a reward model (Eq.5,6).
> The difference is the way to build the dynamics model and the reward model.
> In many traditional world models, the dynamics models and the reward models are built directly by neural networks.
> In contrast, our dynamics model and reward model are built upon memories (the offline dataset) on a metric space learned by neural networks.
> Therefore, we think it is suitable to call VMG a world model.
>
>
> **[Q4] How's the training and inference cost of VMG?**
>
> The training cost is listed in the Appx.D. We train the model for 800 epochs and it takes about 1.5 hours in a single RTX Titan. For inference, building the graph from clustering takes about 0.5-2 minutes before the evaluation. After that, it takes about 0.5-10 minutes to evaluate 100 episodes.
>
> **[Q5] While being able to work in a continuous control is impressive, credit assignment in the transition data graph has been investigated in discrete state/action space in the online RL setting [5,6]**
>
> Thank you for sharing these two papers.
> Indeed, both [5, 6] have studied credit assignments in the graph built from data in the discrete environments, and we extend it to the continuous cases. We will update our related work section to include a discussion in our final version.

---

> > ### Comment · Reviewer_b4KN · 2022-11-18
> > **Response and Further Comments**
> >
> > I would like to thank the authors for the extra experiments that have addressed some of my previous concerns.
> > I do appreciate the gym-locomotion results and statement of the limitation while I'm a bit lost about why VMG perform basically the same as BC. I can imagine the VMG selecting the best high-level path for the action translator to follow (say, for medium expert or medium-replay dataset), eventhough the action translator is suboptimal on the low level.
> > Anyways, it might be difficult to do a thorough study given limited time but given the current empirical results I think It might be better to claim the VMG is for goal-oriented tasks rather than a general offline RL method.

---

> > > ### Author Response · Authors · 2022-11-18
> > > **Reply to Reveiwer b4KN**
> > >
> > > Thank you for your response. The performance metric of these locomotion tasks reflects how fast the agent can move forward. We suppose the performance at the low-level dynamics determines the action quality and thus controls the agent speed even though the agent still follows a good path to move forward. We have updated our abstraction, contribution in the introduction section, and the conclusion section to clarify that the performance advantage of VMG is shown in goal-oriented tasks.

---

### Official Review · Reviewer_UqHS · 2022-10-20

**Confidence:** 4
**Correctness:** 4
**Technical Novelty And Significance:** 3
**Empirical Novelty And Significance:** 4
**Recommendation:** 8

**Clarity, Quality, Novelty And Reproducibility:**

Clarity: Good.
Quality: Good.
Novelty: Good.
Reproducibility: The authors promise to release code. Otherwise, the methodology is clearly explained and should be reproducible.

**Strength And Weaknesses:**

Strength:
* The paper studies a relevant topic (structured abstraction & offline RL).
* The paper is very well written and clear. It does not make things unnecessarily complex, while all necessary details are provided.
* Performance of the methods clearly outperforms the baselines in the tested tasks.
* Good ablation experiments.
* Related work is well covered.
* Good visualizations, such as Figure 4, which really show what the method does.

Weaknesses:
* Sec 3.2: Since each vertex is essentially a D-dimensional ball with radius gamma. These balls can not cover the full space without overlap. What happens if a certain point can be assigned to multiple vertices?
* Sec 3.3: I understand why you construct the combined reward function from internal and transition rewards (with weight ½ on each internal half), but there is a strong assumption here, being that there are as many true steps internal of a vertex as there are between vertices. This will probably not be true in practice: especially with larger values of gamma, the number of steps within a vertex will be much bigger than the number of steps between them. I guess one could argue that you need extra weight on the internal components in the reward function.
* Sec. 3.4: I was highly surprised you used Dijkstra’s, after first having solved the problem with VI. I had written a comment about this, only to find out why you make this choice in the “Multi-step Search” section of the ablation (p 9). Try to explain why you make certain choices directly when you introduce them.
* You only test on three tasks of the D4RL. This is fine, but some variation in environment type would have made the results a bit stronger.


**Summary Of The Paper:**

This paper introduces Value Memory Graphs (VMG), a learned latent MDP structured as a graph. Since the VMG is relatively small, we can apply value iteration in it. The obtained value function is then used in a planner (Dijkstra’s), and the resulting plan in abstract space is converted back to the original space using a learned action decoder. Experiments in several tasks show improved performance over baselines, and investigate the importance of each component through ablation studies.

**Summary Of The Review:**

I recommend this paper for acceptance. It is well-written, covers a relevant topic, has good results, and extensive insight/ablation studies. There are some things to improve, as listed above, but overall I think this would be a valuable addition to the ICLR conference programme.

---

> ### Author Response · Authors · 2022-11-16
> **Reply to Reviewer UqHS**
>
> Thank you for your detailed review, questions, and suggestions. We here address them and will incorporate all the feedback.
>
> **[Q1] Sec 3.2: Since each vertex is essentially a D-dimensional ball with radius gamma. These balls can not cover the full space without overlap. What happens if a certain point can be assigned to multiple vertices?**
>
> Given a state to assign and the vertex set. We will directly compute the distance between the state and all the vertices, and assign the closest vertex to the state. Therefore, each state will only be assigned to one vertex.
>
>
> **[Q2] There is a strong assumption in the reward design, being that there are as many true steps internal of a vertex as there are between vertices.**
>
> Here we design an ablation experiment with a balanced internal reward on the tasks 'kitchen-partial', 'antmaze-medium-play', and 'pen-human'.
> According to our graph construction method described in Sec 3.2, the number of 'true steps' between vertices is always one.
> We checked the average number of 'true steps' in the internal of a vertex in VMGs trained in the tested tasks and listed the numbers below. Let's denote it as $n_{int}$.
> We notice that $n_{int}$ in kitchen-partial and pen-human is equal or close to one under the default hyperparameters we use in the main experiments.
> Then, we define a new variant of the reward function to balance the internal rewards $R_{G, ba}(v_{j_1}, v_{j_2}) = \frac{n_{int}}{2} R_{j_1 \rightarrow j_1} + R_{j_1 \rightarrow j_2} + \frac{n_{int}}{2} R_{j_2 \rightarrow j_2}$,  if $e_{{j_1}\rightarrow{j_2}} \in \mathcal E$.
> Experimental results show that the new reward design performs well in kitchen-partial and pen-human, and slightly worse than the original design (Eq.6) in antmaze-medium-play, suggesting that the internal reward balance is not required with appropriate hyperparameter settings.
>
>
> |               | antmaze-medium-play | kitchen-partial | pen-human |
> |---------------|---------------------|-----------------|-----------|
> | $n_{int}$     | 13.4                | 1.0             | 0.8       |
> | $R_{G, ba}$   | 78.5                | 68.8            | 72.1      |
> | $R\_G (orig)$ | 82.7                | 68.8            | 70.7      |
>
> **[Q3] Try to explain why you make certain choices directly when you introduce them like Dijkstra.**
>
> Thank you for your suggestion. We will rearrange the content to directly explain Dijkstra search in the method section in the final version.
>
>
>
> **[Q4] You only test on three tasks of the D4RL. This is fine, but some variation in environment type would have made the results a bit stronger.**
>
> Here we test VMG on gym-locomotion tasks as also requested by other reviewers.
>
> - **VMG may not help in gym locomotion tasks, as these tasks don't require agents to reason the long future and thus are out of our scope.**
> Gym locomotion tasks provide rich and dense reward signals, and the motion patterns to learn in these tasks are periodic and short.
> Therefore, we think the problems VMG designed to solve are not an issue here.
>
> - **Our performance in these tasks is expected to be close to behavior cloning,** since the low-level component, the action translator, is trained via (conditioned) behavior cloning.
> The action translator is used to handle local dynamics that are not modeled in VMG.
> Here we run new experiments in these tasks and show the results below.
> Experimental results verify our assumption.
>
> - Results and analysis suggest that
> **an improved design and/or learning strategy of the action translator might help improve the performance.**
> For example, training the action translator using conditioned offline RL methods instead of conditioned behavior cloning.
> However, **this is orthogonal to our VMG framework contribution to future reasoning**, and we leave it for future work.
>
> | Dataset                   | VMG  | BC    | CQL   | IQL   |
> |---------------------------|------|-------|-------|-------|
> | halfcheetah-medium        | 42.2 | 42.6  | 44.0  | 47.4  |
> | hopper-medium             | 49.4 | 52.9  | 58.5  | 66.3  |
> | walker2d-medium           | 70.4 | 75.3  | 72.5  | 78.3  |
> | halfcheetah-medium-replay | 38.2 | 36.6  | 45.5  | 44.2  |
> | hopper-medium-replay      | 15.2 | 18.1  | 95.0  | 94.7  |
> | walker2d-medium-replay    | 28.6 | 26.0  | 77.2  | 73.9  |
> | halfcheetah-medium-expert | 80.5 | 55.2  | 91.6  | 86.7  |
> | hopper-medium-expert      | 49.5 | 52.5  | 105.4 | 91.5  |
> | walker2d-medium-expert    | 70.4 | 107.5 | 108.8 | 109.6 |

---

### Official Review · Reviewer_Wbjx · 2022-10-24

**Confidence:** 4
**Correctness:** 3
**Technical Novelty And Significance:** 4
**Empirical Novelty And Significance:** 3
**Recommendation:** 6

**Clarity, Quality, Novelty And Reproducibility:**

The paper is clearly written and easy to follow. I think the authors have done a good job in motivating the problem and introducing the method. The paper is also novel. I think the main limitation is that it is designed for the offline RL setting for now, and is not ready to tackle online RL problems. The authors also try to understand the value memory graph via a reasonable visualization and analysis.

**Strength And Weaknesses:**

### Strengths
- Originality: The authors propose an interesting method to tackle sparse rewards and long-horizon planning, which is different from existing methods (such as improving exploration in sparse reward tasks). Specifically, the paper follows the "abstraction" perspective, which builds an abstract graph of the underlying MDP, which leads to a small state and action spaces that allow for tractable value iteration. To achieve this goal, the authors propose a contrastive learning method for state merging, and also propose an action translator to further translate the abstract action to a real action.

- Clarity: The paper is clearly written and easy to follow. I think the paper motivates the problem well, and describes the procedure clearly including how to build a metric space, how to construct the graph in value memory graph, and how to use such value memory graph.

### Weaknesses
- Quality and significance: My main concern for the paper is the applicability of the method and the significance of the experiments. The paper can be improved by studying the method in the online RL setting (I understand that it is more difficult and challenging to apply the method in online RL settings, but I think that could greatly improve the paper.) In addition, I wonder how the method compare against baselines in commonly-used MuJoCo tasks from D4RL like hopper, walker2d, and halfcheetah. These are tasks with dense rewards, but they also have continuous state and action spaces, and it is worth evaluating VMG in these tasks (since this is also one of the challenges highlighted in the paper). I also have a question in Eq. (2). Why does it need to consider all the other next states? What if sj' and si' are very similar? I will be happy to increase the score if additional experiments on MuJoCo are included and analyzed.

**Summary Of The Paper:**

The main focus of the paper is to tackle some critical challenges in continuous control RL tasks including sparse rewards and long-horizon planning. The idea is to build a simple and discrete world model leveraging an offline dataset (and the paper focuses on the offline RL setting) to learn an abstraction of the original environment. The authors propose Value Memory Graph that represents the MDP as a directed graph with "abstract" state and actions. Then, value iteration can be applied given the small state and action spaces. The authors also learn an action translator to convert abstract actions to real actions later. The authors conduct experiments on some of the tasks from D4RL to evaluate its effectiveness.

**Summary Of The Review:**

The authors propose a novel method to tackle some of the important challenges in DRL. I think the experimental section can be improved by evaluating VMG on commonly-used MuJoCo tasks from D4RL. It can also be improved to extend VMG to online RL problems.

---

> ### Author Response · Authors · 2022-11-15
> **Reply to Reviewer Wbjx**
>
> Thank you for your detailed review, questions, and suggestions. We here address them and will incorporate all the feedback.
>
> **[Q1] The paper can be improved by studying the method in the online RL setting.**
>
> We agree that making VMG work in the online RL setting can be interesting. However, as we target the offline RL setting, the online RL setting is out of our scope and may require additional designs on exploration. We leave it for future work.
>
> **[Q2] I wonder how the method compares against baselines in commonly-used MuJoCo tasks from D4RL like hopper, walker2d, and halfcheetah.**
>
> - **VMG is introduced to help agents reason the long future better** so as to improve their performance in complex environments with sparse rewards and large search space due to long temporal horizons and continuous state/action spaces.
>
> - **VMG may not help in gym locomotion tasks, as these tasks don't require agents to reason the long future and thus are out of our scope.**
> Gym locomotion tasks provide rich and dense reward signals, and the motion patterns to learn in these tasks are periodic and short.
> Therefore, we think the problems VMG designed to solve are not an issue here.
>
> - **Our performance in these tasks is expected to be close to behavior cloning,** since the low-level component, the action translator, is trained via (conditioned) behavior cloning.
> The action translator is used to handle local dynamics that are not modeled in VMG.
> Here we run new experiments in these tasks and show the results below.
> Experimental results verify our assumption.
>
> - Results and analysis suggest that
> **an improved design and/or learning strategy of the action translator might help improve the performance.**
> For example, training the action translator using conditioned offline RL methods instead of conditioned behavior cloning.
> However, **this is orthogonal to our VMG framework contribution to future reasoning**, and we leave it for future work.
>
> | Dataset                   | VMG  | BC    | CQL   | IQL   |
> |---------------------------|------|-------|-------|-------|
> | halfcheetah-medium        | 42.2 | 42.6  | 44.0  | 47.4  |
> | hopper-medium             | 49.4 | 52.9  | 58.5  | 66.3  |
> | walker2d-medium           | 70.4 | 75.3  | 72.5  | 78.3  |
> | halfcheetah-medium-replay | 38.2 | 36.6  | 45.5  | 44.2  |
> | hopper-medium-replay      | 15.2 | 18.1  | 95.0  | 94.7  |
> | walker2d-medium-replay    | 28.6 | 26.0  | 77.2  | 73.9  |
> | halfcheetah-medium-expert | 80.5 | 55.2  | 91.6  | 86.7  |
> | hopper-medium-expert      | 49.5 | 52.5  | 105.4 | 91.5  |
> | walker2d-medium-expert    | 70.4 | 107.5 | 108.8 | 109.6 |
>
>
> **[Q3] Eq. (2). Why does it need to consider all the other next states in the same batch? What if sj' and si' are very similar?**
>
> This design follows the common practice in many contrastive learning methods like CLIP [1].
> It is possible that the negative samples can be similar in some rare cases.
> However, as the batch of transitions is uniformly and randomly sampled from the dataset, the probability to obtain such false negative samples is low compared to the dissimilar samples, and thus its influence on metric space learning is limited.
>
> [1] Radford, Alec, et al. "Learning transferable visual models from natural language supervision." International Conference on Machine Learning. PMLR, 2021.

---

### Official Review · Reviewer_ZzyC · 2022-10-25

**Confidence:** 5
**Correctness:** 2
**Technical Novelty And Significance:** 2
**Empirical Novelty And Significance:** 3
**Recommendation:** 6

**Clarity, Quality, Novelty And Reproducibility:**

-

**Strength And Weaknesses:**

While the empirical results in isolation is very convincing, the paper lacks rigor in different design choices of the algorithm. The paper will definitely provide more value to the reader with better explanation on the design choices along with more discussion on the implementation details as outlined below.  would be a good contribution  VI and leveraging the graph for planning in continuous domain.

Strengths:

- The paper definitely boasts some empirical novelty as optimal planning on abstract graphs have not been explored well for d4rl tasks. The approach discussed performs well for the  shown subset of these d4rl tasks.

- The ablation studies presented does hint towards the contribution of different loss functions used towards representation learning / metric learning which is employed by the graph building algorithm.

- The notion of building a abstract graph for exact planning and combining it with lookahead policy (employing goal conditioned inverse dynamics) is powerful and broad in its scope


Weakness:

- The design choices made throughout the approach lacks some rigorous justification as no theoretical evidence has been provided for these choices.

    - [C1]  (eq 1)Why was m set to 1 ? this value will definitely have bunch implications depending on the dimensionality of the latent space and the complexity of the environment dynamics.

    - [C2] (eq 2) why would we take max of the l2 norm and M. could we not have replaced this with max(l2(norm)-m, 0). In the current formulation this component of the loss will never go to zero if I understand it correctly.

    - [C3] (eq 4) We are basically ignoring any “self loops” that arise with this kind of clustering and only considering dynamics that take us from one state to another. Why is this? How often do these self loops occur. Moreover, the reward design from vj1 to vj2 seems arbitrary and not backed by any theory. Here we are basically assuming uniform weighting over all states in a cluster. I understand that adding half of the rewards from both self loops in source and target state, allows the trajectory reward to sum up to the true trajectory reward, but the reward definition here does seem arbitrary, which is hard to justify with empirical evaluations alone.

    - [C4] I did not quite get the notion of using Dijkstra’s search to find the best path across the world model, using one more weighting mechanism that is not theoretically justified. Would the Value iteration already not provide a relatively optimal path from any state to the best state reachable ?

    - [C5] The choice of k = 10 (for inverse dynamics) seems arbitrary as we are only using 1 or 2 step lookahead goals for the models. It seems that we are baking more into the prior as 1 or 2 step lookahead in the abstract space can technically be much more than 1 or 2 as we are aggregating states to a single cluster. i.e. a transition in abstract space may amount to more than one step in the real world, how much of this is actually happening in empirical evaluations is unclear.

    - [C6] The choices of penalty beta (from table 6 in appendix) seems low and arbitrary. generally we set this to 0.99. It does make sense that we would like to have smaller discount factors in abstract models as well as constrain it further to reduce cumulative errors, however the choices made here even without a hyper-parameter search is not rigorous by any means.

- Lacks some key implementation details.

    - What was the dimension of the latent space used for the latent dynamics model.

    - No training curves has been presented.

    - When we mention ““best one from the checkpoints saved from the 500th to the 800th epochs.” ([pdf](zotero://open-pdf/library/items/NHHMIX9V?page=15)) does it mean best in terms of the training loss / validation losses?

    - The values used for hyper-parameter search are not listed down, only the best performing ones. Moreover the number of runs used for hyper-parameter search must be mentioned in the main paper for clarity, as it is one of the crucial yet generally overseen.

    - Again the choices made for these hyper-parameter sweeps does not seem to be well justified.

- The paper is missing some key connections in the related work section.

    - Bisimulation metrics has been well explored for learning abstract models for a long time now. [4, 5, 6]. They provide a rigorous frame  and other abstraction literature is not cited well.

    - Secondly optimal planning on graph structured world models has also been explored in works like [1,2,3]. [1] Here is especially relevant as the world model is constructed using metric learned in abstract space. Careful comparison / discussion of these related works would greatly bolster the context of the paper.


[1] Shrestha, A., Lee, S., Tadepalli, P., & Fern, A. (2021). DeepAveragers: Offline Reinforcement Learning by Solving Derived Non-Parametric MDPs. *ArXiv, abs/2010.08891*.

[2] Char, I., Mehta, V., Villaflor, A., Dolan, J.M., & Schneider, J.G. (2022). BATS: Best Action Trajectory Stitching. *ArXiv, abs/2204.12026*.

[3] Marklund, H., Nair, S., & Finn, C. (2020). Exact (Then Approximate) Dynamic Programming for Deep Reinforcement Learning.

[4] Ferns, N., & Precup, D. (2014). Bisimulation Metrics are Optimal Value Functions. *UAI*.

[5] Ferns, N., Panangaden, P., & Precup, D. (2011). Bisimulation Metrics for Continuous Markov Decision Processes. *SIAM J. Comput., 40*, 1662-1714.

[6] Bertsekas, D.P. (2009). Neuro-Dynamic Programming. *Encyclopedia of Optimization*.

**Summary Of The Paper:**

The paper proposes learning a graph structured abstract world model for offline reinforcement learning. Here the learned abstract world model is built on top of a learned representation and used in conjunction with Value Iteration solver, Dijkstra’s search, and temporal inverse dynamics model. In general learned abstract models are powerful when combined with optimal planning and look-ahead policies. Here the authors use a clustering based approach to build the world model and show empirical evaluation of such an approach on a subset of D4RL tasks. Here the empirical evaluation is outperforms the current SOTA methods for the selective domains shown in the paper.

**Summary Of The Review:**

In the current state of the paper, I will favor rejection. however I look forward to the authors rebuttal on my questions / concerns and am open to updating the score afterwards.

---

> ### Author Response · Authors · 2022-11-14
> **Reply To Reviewer ZzyC Part 1**
>
> Thank you for your detailed review, questions, and suggestions. We here address them and will incorporate all the feedback.
>
> **[Q1] (Eq 1)Why was m set to 1? This value will definitely have a bunch of implications depending on the dimensionality of the latent space and the complexity of the environment dynamics.**
>
> We tried m=1 at first and noticed a reasonable performance in our early experiments.
> As we try to avoid a dense search on hyperparameters, we didn't tune m later.
> We agree that m can have a bunch of implications depending on different tasks.
> Here we show new ablation studies with m=0.5, 1, and 2 in the datasets antmaze-medium-play, kitchen-partial, and pen-human.
> In the antmaze experiment, performance becomes better with a larger m. But in the task pen-human, a smaller m gives us better results.
> In kitchen-partial, m=1 shows the best performance.
> Experimental results suggest that m=1 is a reasonable value for the tasks we evaluate on.
> And if we tune m separately, it is possible to improve the performance.
>
> |          m          |  0.5 |   1  |   2  |
> |:-------------------:|:----:|:----:|:----:|
> | antmaze-medium-play | 65.0 | 82.7 | 84.0 |
> |   kitchen-partial   | 17.0 | 68.8 | 64.5 |
> |      pen-human      | 75.1 | 70.7 | 65.7 |
>
> **[Q2] (eq 2) Why would we take max of the l2 norm and M. Could we not have replaced this with max(l2(norm)-m, 0).**
>
> Yes, the new format equals the old format minus a constant, $max(l2(norm)-m, 0) = max(l2(norm), m) - m$. Therefore, both formats have exactly the same gradient. We will update Eq.2 as the new format so the minimal value of the loss will be zero.
>
> **[Q3] The reward design from vj1 to vj2 seems arbitrary and not backed by any theory. Here we are basically assuming uniform weighting over all states in a cluster. The reward design is hard to justify with empirical evaluations alone.**
>
> Regarding the uniform weighting, as each transition in the offline dataset can be viewed as a sample of the transition distribution under the (unknown) behavior policy, we think calculating internal rewards via averaging over all states in a cluster with uniform weights can be viewed as an unbiased estimation of the real internal reward under the behavior policy.
> In our ablation study (Sec.4,4, Paragraph 'Graph Reward'), we show the performance of six different reward designs.
> Experimental results in Tab.3 show that our reward design performs better than the other five reward design candidates and can give us reasonable performance in the tested tasks, which supports our choice.
>
> **[Q4] The choice of K = 10 (for inverse dynamics) seems arbitrary as we are only using 1 or 2-step lookahead goals for the models. It seems that a transition in abstract space may amount to more than one step in the real world.**
>
> The one-step transition in the graph corresponds to multi-steps in the environment.
> Therefore, K should be larger than 1 or 2.
> We checked the **average** numbers of environment transitions in each graph transition in our main experiments and show three tasks in the following table.
> Numbers of the remaining tasks will be included in the paper.
>
> |                      | antmaze-medium-play | kitchen-partial | pen-human |
> |:--------------------:|:-------------------:|:---------------:|:---------:|
> | avg. env. transition |         14.4        |       2.0       |    1.8    |
>
> K=10 was the first value we tried, and we noticed a reasonable performance in the early experiments.
> Therefore, we didn't tune it later for all our experiments to avoid extensive hyperparameter searches.
> To study the influence of K, here we show new ablation experiments with k=5, 10, and 20.
> We notice that K=5 doesn't work in antmaze-medium and kitchen-partial, which suggests that K=5 is not big enough to cover 2 steps in the graph transition.
> In addition, the experiments with K=20 show better results than K=10 in kitchen-partial and pen-human.
> In antmaze-medium-play, K=10 performs the best.
> Experimental results suggest that a big enough K helps the model perform better.
>
> |          K          |   5  |  10  |  20  |
> |:-------------------:|:----:|:----:|:----:|
> | antmaze-medium-play |  7.0 | 82.7 | 74.0 |
> |   kitchen-partial   |  0.3 | 68.8 | 76.8 |
> |      pen-human      | 80.3 | 70.7 | 83.5 |

---

> ### Author Response · Authors · 2022-11-14
> **Reply to Reviewer ZzyC Part 2**
>
> **[Q5] (Eq 4) We are basically ignoring any “self loops” that arise with this kind of clustering and only considering dynamics that take us from one state to another. Why is this? How often do these self-loops occur?**
>
> We checked all the datasets we use and found no 'self-loop' transition in these datasets.
> This is expected as our tested environments have continuous and high-dimension state spaces, and the agent is unlike to return exactly the same state after executing an action.
> Besides, VMG is an abstract of the environment not only from the state/action space perspective but also from the time perspective.
> A transition in VMG corresponds to multiple transitions in the environment as explained in the last question.
> Therefore, for an environment transition in which both the current state and the next state are classified to the same vertex, we don't view it as a (self-loop) VMG transition, but as a subpart of a VMG transition among nodes.
> In addition, a self-loop transition in our test environments basically means the agent is stuck in some states and does nothing meaningful.
> Such a stuck does not provide useful information for future reasoning.
>
> **[Q6] The choices of penalty beta from Tab.6 seem low and arbitrary. Generally, we set this to 0.99. It does make sense that we would like to have smaller discount factors in abstract models as well as constrain it further to reduce cumulative errors. However, the choices made here even without a hyper-parameter search are not rigorous by any means.**
>
> We roughly checked the performance with beta=0.8 and 0.95 in the early experiments and selected 0.95 for the kitchen and 0.8 for the remaining domains.
> Here we show new ablation studies with beta=0.8, 0.95, and 0.99.
> Experiments show that beta=0.99 leads to better performance in pen-human and comparable performance in kitchen-partial.
> In antmaze-medium-play, 0.99 performs worse than 0.8 and 0.95.
> We agree with Reviewer ZzyC that a small beta in antmaze might help reduce cumulative errors.
>
> |       beta       |  0.8 | 0.95 | 0.99 |
> |:-------------------:|:----:|:----:|:----:|
> | antmaze-medium-play | 82.7 | 76.3 | 75.0 |
> |   kitchen-partial   | 58.2 | 68.8 | 68.1 |
> |      pen-human      | 70.7 | 69.0 | 74.8 |
>
>
> **[Q7] What was the dimension of the latent space used for the latent dynamics model.**
>
> We set it as 10 at first and noticed a reasonable performance in our early experiments.
> As we try to avoid a dense search on hyperparameters, we didn't tune it later.
> To study the influence of the latent space dimension, we run new ablation experiments here with dimension sizes 5, 10, and 20.
> Experimental results show that models with latent space dimensions 10 and 20 perform better than those with 5, which suggests that a reasonable performance requires big enough dimensions of the latent space to represent the states and actions better.
> Besides, space with 10 dimensions works better than 20 in kithcen-partial but worse than 20 in pen-human, this suggests the performance has space to improve if we tune the dimensions individually in each task.
>
> |      space dim      |   5  |  10  |  20  |
> |:-------------------:|:----:|:----:|:----:|
> | antmaze-medium-play | 71.0 | 82.7 | 82.0 |
> |   kitchen-partial   | 5.75 | 68.8 | 46.0 |
> |      pen-human      | 70.7 | 70.7 | 79.0 |
>
>
> **[Q8] How do you pick the ''best one from the checkpoints saved from the 500th to the 800th epochs.''?**
>
> Due to a lack of a proper validation error metric, deciding the training epochs has been an open problem in offline RL [1,2].
> We select the best checkpoints by evaluating models online in the environment following existing methods like ATAC [3]. Tuning the hyperparameters offline is an ongoing and important research topic in offline RL, and we left it for future work.
>
> [1] Kumar, Aviral, et al. "Conservative q-learning for offline reinforcement learning." Advances in Neural Information Processing Systems 33 (2020): 1179-1191.
>
> [2] Levine, Sergey, et al. "Offline reinforcement learning: Tutorial, review, and perspectives on open problems." arXiv preprint arXiv:2005.01643 (2020).
>
> [3] Cheng, Ching-An, et al. "Adversarially trained actor critic for offline reinforcement learning." Proceedings of the 39th International Conference on Machine Learning (2022).

---

> ### Author Response · Authors · 2022-11-14
> **Reply to Reviewer ZzyC Part 3**
>
> **[Q9] The values used for hyper-parameter search are not listed down, only the best-performing ones. Moreover, the number of runs used for hyper-parameter search must be mentioned in the main paper for clarity, as it is one of the crucial yet generally overseen.**
>
> We searched four hyperparameters in our main experiments: $\gamma_m$ in [0.3, 0.5, 0.8, 1.0, 1.2], beta (reward discount) in [0.8, 0.95], $N_{sg}$ in [1, 2, 3], $N_s$ in [12, $\infty$].
> Hyperparameters are searched one by one, in total 12 configurations. In the main experiments, hyperparameters were searched in 'antmaze-umaze-diverse' for AntMaze domain, 'kitchen-partial' for Kitchen domain. For Adroit, hyperparameters are searched individually.
> We will update the main paper to contain the searched value information.
>
> **The reply to the remaining questions will be posted and the paper will be updated to include new discussions soon**

---

> ### Author Response · Authors · 2022-11-18
> **Reply to Reviewer ZzyC Part 4**
>
> **[Q10] Bisimulation metrics have been well explored for learning abstract models for a long time now. [4, 5, 6]. They provide a rigorous frame and other abstraction literature is not cited well. Secondly, optimal planning on graph structured world models has also been explored in works like [1,2,3].**
>
>
> We have updated our related work section to include a discussion on representation learning including contrastive learning and bisimulation metrics.
> For graph-structured world models, [1, 3] use kNN to discretize the continuous state space. However, they are designed for tasks with discrete actions only and thus cannot be applied in tasks with continuous action spaces like robotics. [2] creates graphs by learning to add new transitions to the dataset. [2] can work in simple tasks with low-dimensional continuous action space like Continous Moutain Car (1 dimension) and Maze2D (2 dimensions), but it is hard to scale to more complex tasks with high-dimensional action space.
> VMG is a method designed for continuous action space. VMG is tested in tasks with up to 24 action dimensions  (like the task 'pen-human' ).
> In addition, although [1] also clusters the states in an abstract space, [1] doesn't propose methods to learn this abstract space. Instead, it directly uses intermediate outputs of pretrained RL models as the state feature for the clustering.
> In contrast, we propose a contrastive-learning-based method to learn our metric space.
> We have updated our related work section to include a discussion for [1,2,3].
>
>
> **[Q11] I did not quite get the notion of using Dijkstra’s search to find the best path across the world model. Would the Value iteration already not provide a relatively optimal path from any state to the best state reachable?**
>
> Our earlier experiments show better performance if we search for the best state first and then plan a path to it.
> Therefore, we keep it in our method.
> We think the reason is the accumulated errors caused by the gap between the VMG and the environment.
> VMG is learned from an offline dataset collected by unknown behavior policies.
> Due to the distribution shift between the offline dataset and the online environment, there can be gaps between the offline learned VMG and the environment.
> The path from the current state to the best reachable state given directly by value iteration (denoted as $P_{VI}$) will be definitely the optimal path on the graph-MDP. However, a path searched by Dijkstra (denoted as $P_D$) may be shorter on the graph.
> Due to the gap between VMG and the environment, an optimal path on VMG may not be still optimal in the environment. At the same time, a shorter path helps reduce cumulative errors and uncertainty, and thus increases the reliability of the policy.
> The weighting mechanism is a trade-off between $P_{VI}$ and $P_D$. We use the gap between the state value and the maximal value as the weights. As Dijkstra searches for a path with minimal accumulated weights,
> the searched path under the weighting mechanism is encouraged to be relatively short and also high-value in each transition.
> Our ablation study in Tab.4 of the main paper shows that this multi-step search mechanism improves the model performance, which verifies our design choice.
>
> **[Q12] No training curves have been presented.**
>
> Training curves now are included in Appx.F in the updated version.

---

> ### Comment · Reviewer_ZzyC · 2022-11-23
> **Updated Scores.**
>
> I still have some reservations regarding if the approach outlined in the paper is a "principled" one, For example most ablation studies shows that the evaluations in kitchen-partial is very sensitive when it comes to the hyper parameter choices. Moreover, the appraoch does not tranfer well to locomtion tasks such as Halfcheetah and may need some more cusomized solutions.
>
> Nevertheless, the empirical results shown is definitely compelling enough for the community to learn from the design choices that has been made along the way, albeit I am still not convinced that all the design choices made here were correct. The update in conclusion section to clarify that the performance advantage of VMG is shown in goal-oriented tasks was very welcome.
>
> sidenote: if the different epochs were used for evaluation, these must be counted as hyperparamters and the number of hyperparamter tests, i.e. Number of evaluation iterations must be updated to include the search for "best epoch". I would be curious if this will now be 12*3 or 12+3?

---

> > ### Author Response · Authors · 2022-11-28
> > **Reply to Reviewer ZzyC**
> >
> > Thank you for your response. We agree that our design has space to improve. However, we believe VMG shows the value of building a simple and discrete world model to represent the original complex and continuous environment. Albeit our concrete design and implementation limitations, we hope VMG can inspire more future works to solve tasks with reasoning challenges.
> > We will update the number of evaluation iterations to include the checkpoint searching in the final version. Other hyperparameters are searched under the same checkpoint. Therefore, it is a 'sum' instead of a 'multiplication'.

---

### Author Response · Authors · 2022-11-18
**General Response and Updated Manuscript**

We thank the reviewers for their insightful feedback. It encouraged us that
our technical novelty (Wbjx, UqHS, b4KN) and empirical novelty (ZzyC, Wbjx, UqHS, b4KN) are significant;
our method is powerful and broad in its scope (ZzyC), interesting (Wbjx), and may have further impacts (b4KN);
our performance is convincing (ZzyC) and good in the tested tasks (UqHS, b4KN)
our writing is clear (Wbjx, UqHS, b4KN) with good visualization (UqHS, b4KN).
We address the reviewer’s comments below and will incorporate all the feedback and discussions into the final version.
In addition, the manuscript is updated. The modified parts are colored blue.
The main changes include a discussion in representation learning and graph structure in related work, the new ablation experiments in the appendix, more information about the hyperparameter in the experiments section and the appendix, a discussion about gym locomotion in the limitation section and the appendix, and learning curves in the appendix.

---

### Public Comment · ~Wenyan_Yang1 · 2023-03-03
**The reproducibility results of Table**

I was using the supplementary code to reproduce the offline planning results for kitchen tasks. However, the performance results have a huge gap between my results and the one in Table.1.

Currently, I only reproduced the “kitchen-complete” and "kitchen-partial," so the performances I got are: a) kitchen complete  43.78 (12.3) (reproduced) and for kitchen partial  I have 2.3 (11.3)

I strictly followed the instructions for installation and did not change the default parameters. I wonder if there are any hyperparameters that I should pay attention to. Could you provide details about the configuration of the code so I can reproduce similar results?

---

> ### Author Response · Authors · 2023-03-03
> **Reply**
>
> Thank you for your interest in our work! The code in the supplementary is not the final release and missing details about training and evaluation. We are still cleaning and preparing the code and will open-source it in our GitHub repository soon once it is done. Before the code releasing, you can find the pretrained models and an instruction to evaluate them for the two tasks you have tried kitchen-complete and kitchen-partial [here](https://drive.google.com/file/d/1I2CS18gtottmORI_mExdIlv-XEMSFUIx/view?usp=share_link). Please let me know if you have further questions. Thank you!

---

### Decision · Program_Chairs · 2023-01-20

**Decision:**

Accept: poster

**Justification For Why Not Higher Score:**

There are concerns remaining about the complexity of the method, that it is difficult to apply to a wider selection of environments, and lacks a rigorous foundation.

**Justification For Why Not Lower Score:**

In spite of concerns, the reviewers agreed the work was interesting and the results presented were compelling. While building on top of the method as a whole may be difficult there are insights in the paper from which future work may nonetheless take inspiration.


**Metareview: Summary, Strengths And Weaknesses:**

This paper proposes learning a graph based world model using offline RL data for efficient exploration and planning. The key motivation is to use this framework to solve sparse reward and long horizon planning problems. A graph is constructed using contrastive losses and then used in conjunction with value iteration on the graph + an inverse dynamics action model for behavior generation.

The graph is constructed by clustering together experiences in a node. The full environment can then be represented as a set of these nodes and links between them. It seems like the novelty here is combine existing ideas of value iteration, cluster based graphs and inverse dynamics model on a particular set of sparse reward RL environments. The ablations seem sound and reasonable. There was an agreement amongst reviewers that the combination of the graph MDP construction with inverse dynamics model is novel and interesting.

The reason why this paper did not receive stronger reviews or a strong consensus is because: 1) the design choices made have some empirical backing but largely unexplained and hard to contextualize for other RL problems, 2) the paper does very little in terms of providing a solid mathematical/rigorous foundation for graph based value iteration + inverse dynamics approach, 3) the experiments are limited to a particular set of behaviors/environments and it is not clear if the results will hold for other environments.

**Note From Pc:**

if the above contains the word "oral" or "spotlight" please see: "oral" presentation means -> notable-top-5% and "spotlight" means -> notable-top-25%. As stated in our emails, we are disassociating presentation type from AC recommendations

**Summary Of Ac-Reviewer Meeting:**

1. Before the rebuttal it should not be accepted as is. But the authors changed a few things after the rebuttal. It is definitely an interesting paper but it is not a foundation paper yet that people would build on it. They would read it to understand the implementation details but it is not clear if it is a strong enough paper to build on top of.

2. In offline RL the main bottleneck was how many evaluations in the online env should we use to get a good average. The authors were not stringent about this. They have a few hyper param test cases and they do this for 3 epochs. The number they put out is good — I appreciate those numbers. Architecture makes sense. It is not like anyone can do better in this case. They don’t need to beat SOTA — results are impressive enough that people should look at this. Some details like inverse action generator and value iteration within this framework is valuable — generally overlooked in offline RL setting.

3. It does not have a rigorous foundation — other people can then build on top of it. It does not provide a theoretical foundation. Their version is to build the graph, do value iteration and then we add a bunch of bells/whistles (e.g. dijkstra’s — why? not clear. Why value iteration not good enough? No answer. Why inverse action simulator? Open question). So there are a lot of open questions. There are no PAC guarantees because they don’t formulate or structure the graph. Idea is good but it needs a rigorous approach if someone needs to build on top of it.

4. They found something that works for these specific benchmarks for goal-oriented settings (not general off policy). It is not clear how this can be replicated to other domains. It is interesting enough to see but not sure if its strong enough to be accepted.

5. Would not lean on rejection. It is probably worth accepting if all clarifications are made.